# The Wolf King's Leisure Estate: An Andalusi Agricultural and Palatine Project (Murcia, 12th Century)

Julio Navarro-Palazón * and Pedro Jiménez-Castillo *

Spanish National Research Council, Laboratorio de Arqueología y Arquitectura de la Ciudad,
18010 Granada, Spain
* Correspondence: julionavarro@eea.csic.es (J.N.-P.); pedro@eea.csic.es (P.J.-C.)

**Abstract:** The Castillejo de Monteagudo, which has been well known since excavations began in 1924, is a palatial residence built on a promontory. However, the fact that it was part of an extensive agricultural estate, known as Ḥiṣn al-Faraj, which included dry-farming, orchards, gardens, woodland, hunting areas, and marshes, as well as important hydraulic infrastructures, has not been sufficiently emphasised to date. Archaeological research on the irrigated plain during 2018 and 2019 has brought to light part of the palatine area, which was organised around a large garden presided by a residential complex with a porticoed pavilion and a pool at the centre. All known buildings date to the reign of Emir Ibn Mardanīš (1147–1171), although the possibility that the estate was created earlier cannot be ruled out. It was destroyed twice by the Almohads (1165 and 1171) and reused by the Castilian King Alfonso X, perhaps after being restored by Ibn Hūd al-Mutawakkil (1228–1238).

**Keywords:** Islamic palatine architecture; royal country estate; al-Andalus; Ibn Mardanīš; Castillejo de Monteagudo; 12th century

## 1. Introduction

The royal country estate (sp. *almunia*; ar. *munya*) of Castillejo de Monteagudo is located 5 km to the northeast of the city of Murcia (Figure 1), on the edge of the *huerta* irrigated by the Segura River (Figure 2).[1] In the Middle Ages, the irrigated lands of the alluvial plain reached the large areas dedicated to dryland agriculture and woodland, situated on the low foothills to the north. The area was crisscrossed by wadis (*ramblas*), which flowed into the extensive marshes below; the marshes, which were drained by ditches and turned into orchards, no longer exist, but are abundantly attested in the written sources. Consequently, the Castillejo was a means to colonise barren lands near the city while remaining close to areas of woodland and marshland, used as hunting grounds for the inhabitants of the royal estate.

The fortified palace of Castillejo was excavated by Andrés Sobejano in 1924–1925 and has not been the subject of archaeological excavations since (Figures 3 and 4); however, it has produced an abundant bibliography because, for many years, it was one of the few known 12th-century palaces in the Muslim West.[2] Built by Amir Abū 'Abd Allāh Muḥammad b. Sa'd b. Mardanīš al-Juḏāmī, the Wolf King of the Christian chronicles, the palace has often been mentioned as part of the building programme undertaken by this historical figure, which mainly took place in the city of Murcia.[3]

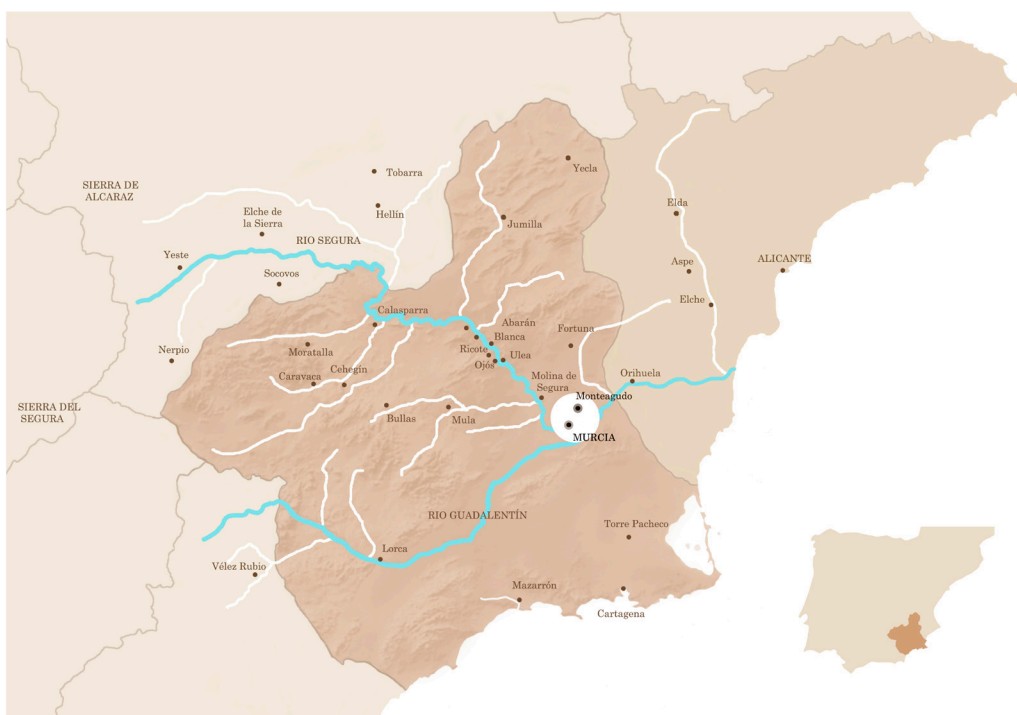

**Figure 1.** Location of the city of Murcia and Monteagudo in the southeast of the Iberian Peninsula. In darker colour, the territory of the modern region of Murcia.

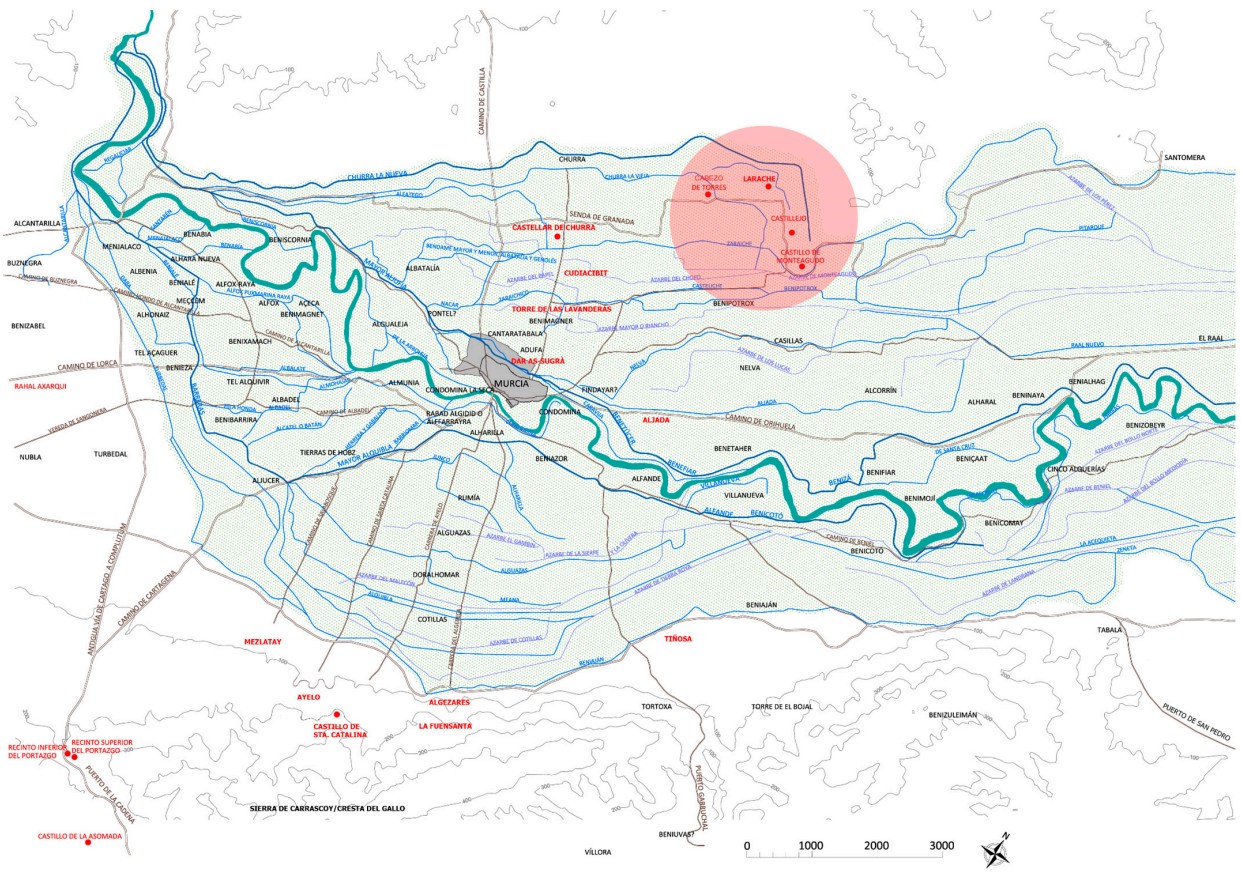

**Figure 2.** Map of the *huerta* of Murcia in the 13th century. The place names mentioned in the text are highlighted in red.

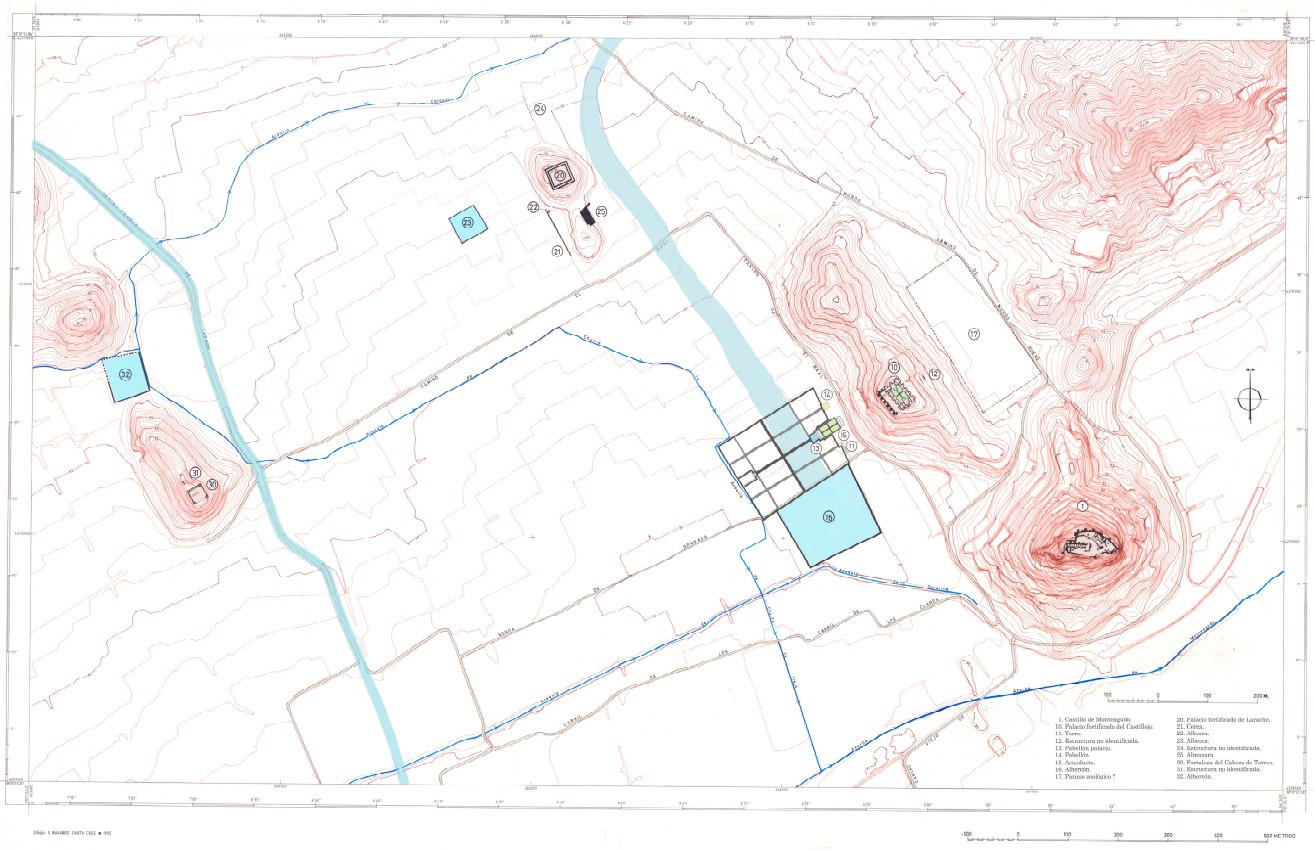

**Figure 3.** Location of the Castillejo de Monteagudo estate on the municipal map of Murcia.

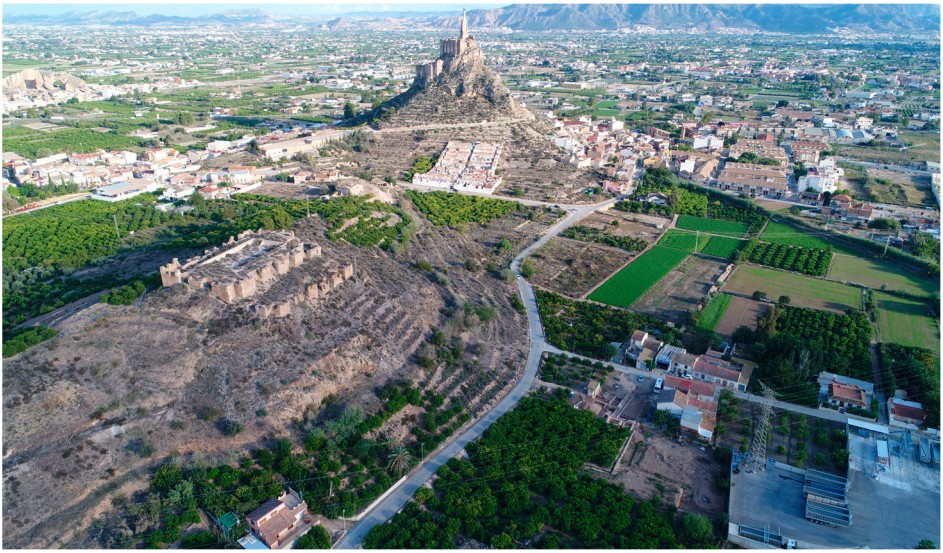

**Figure 4.** General view of the Castillejo from the north; behind, the Monteagudo Castle.

The earliest news of the existence of other architectural remains near the fortified palace dates back to Lozano's work *Batistania y Contestania del Reino de Murcia* (Lozano 1794, I, pp. 160–71). The text mentions the existence of a major hydraulic work: "*The great aqueduct, of strong mortar, runs through the Olivar and crosses the vicinity of the town. The farmers say that it can be seen near La Ñora*". Between 1905 and 1907, González Simancas wrote the *Catálogo Monumental de la Provincia de Murcia*, a widely-consulted manuscript, although it was not published until 1997, that describes some of these remains:

"In the plain, to the SE and near the castle, the foundations of a mortar wall have recently been discovered while ploughing; the wall follows a straight line from east to west for a distance of 58 m, but it must be much longer as it seems to continue under the nearby uncultivated fields. Given its layout, this could well be a remnant of the 'large aqueduct of firm mortar' that Lozano saw in the late 18th century in an olive grove near the village".

The study of the archaeological complex began with the excavation of the Monteagudo Castle (1983–1990),[4] followed in 1989 by the survey of the orchards between Cabezo de Torres and Monteagudo. The results of the survey were published in 1993, and they showed that the Castillejo palace, and a series of associated architectural features, were located in the middle of an extensive agricultural estate (Navarro Palazón and Jiménez Castillo 1993). Shortly afterwards, José Manzano also explored the area, reporting his findings in two publications in 1998 (pp. 414–16) and 2007. Finally, Almela Legorburu's (2015) recent master's thesis examined the large agricultural estates associated with the *almunias* of Monteagudo and Cabezo de Torres.

In this article, in addition to publishing recent data on the estate as a whole (Figure 3), we aim to present the palatine complex excavated on the plain, to the west of Castillejo (Figure 4), in 2018 and 2019. The excavation, together with the detailed analysis of the plots of land that surrounded the excavated remains, carried out with the aid of historical aerial photographs, allowed us to define a large palatine area organised around a transept garden, which includes both the recently discovered archaeological remains and those known since the early 20th century.

As we shall see below, Castilian texts report that the estate was unusually large. In addition to the usual irrigated fields, it included large expanses dedicated to dryland agriculture, woodland, and marshland, which makes it unlikely that the whole was surrounded by a single wall (Figure 3). What is certain is that the palatine areas, at least, were surrounded by a wall, although no remains of this wall have been found to date. Considering the size of the estate and its uncertain boundaries, we have come to the conclusion that both the marshland to the south and the Rambla del Carmen to the east were the estate's natural boundaries. No evidence exists concerning the estate's northern and western boundaries, but we believe that large dryland agricultural areas, which were irrigated by, at least, the Caracol and the Carmen ravines, existed in these directions.

According to this hypothesis, the area around the Larache castle (Figures 5 and 6), which is 350 m to the northwest of the Castillejo, was part of the estate. Larache is a roughly quadrangular building without towers, 2160 m$^2$ in size; the perimeter comprises two concentric walled enclosures, 2 m apart, so the exterior wall should be interpreted as an *antemurium*. At the time, we interpreted it as the fortified residence of an Andalusian *almunia* dated to the mid-12th century (Navarro Palazón and Jiménez Castillo 1993, p. 449), and thus separate from the Castillejo, but we are currently inclined to believe that both buildings were part of the same royal estate. As the excavations carried out in 2004 (López Martínez et al. 2005) and (Pujante Martínez and García Ruiz 2008) did not yield conclusive data, some uncertainty exists concerning its chronology, although the two most plausible hypotheses, in our opinion, are: (a) it was built anew during the Hudi period (third *taifas*), as originally argued by José Antonio Manzano (Manzano Martínez and Bernal Pascual 1992, pp. 165–66; Manzano Martínez 1998, p. 430); (b) it was built during the Christian period. The former hypothesis is supported by the information provided by another Murcian *almunia*, Dār aṣ-Ṣuġrà/Qaṣr al- Ṣaġīr, in which the old palace of Ibn Mardanīš was replaced by a more recent complex that we attribute to Ibn Hūd al-Mutawakkil (1228–1238) (Navarro Palazón and Jiménez Castillo 2011a, 2012). The need for all newly-established rulers to express their legitimacy and power through architecture is a widespread and well-studied phenomenon in medieval Islam, which may have also been at play at the Monteagudo estate.[5] The second hypothesis is based on the fact that the 'Real de la Reyna' of Monteagudo was granted to the Church of Cartagena in 1311, together with the site of Alguazas;[6] the donation was made effective in 1321. Bishop Peñaranda (1327–1351)

built the so-called "Torre Vieja" (Old Tower) in Alguazas in the mid-14th century, so it is possible that he did the same in Larache.[7] Against this hypothesis is the fact that late medieval written sources, including the *Fundamentum Ecclesiae Carthaginensis*, by Bishop Diego Comontes, which lists the works of Bishop Peñaranda in detail, say nothing about Church constructions in Larache.

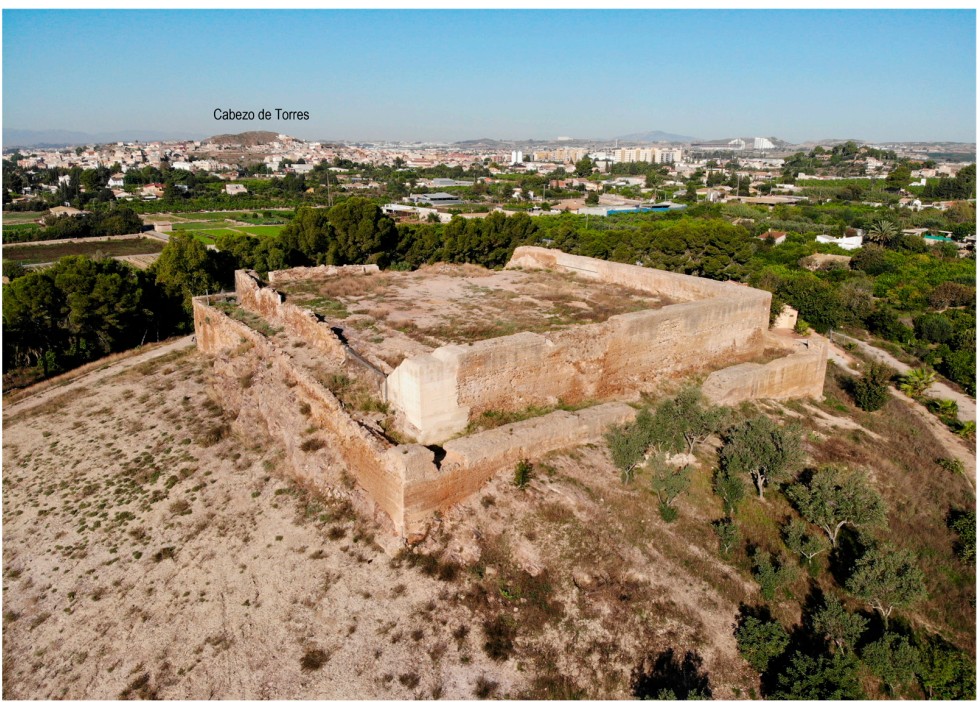

**Figure 5.** Larache castle seen from the southeast.

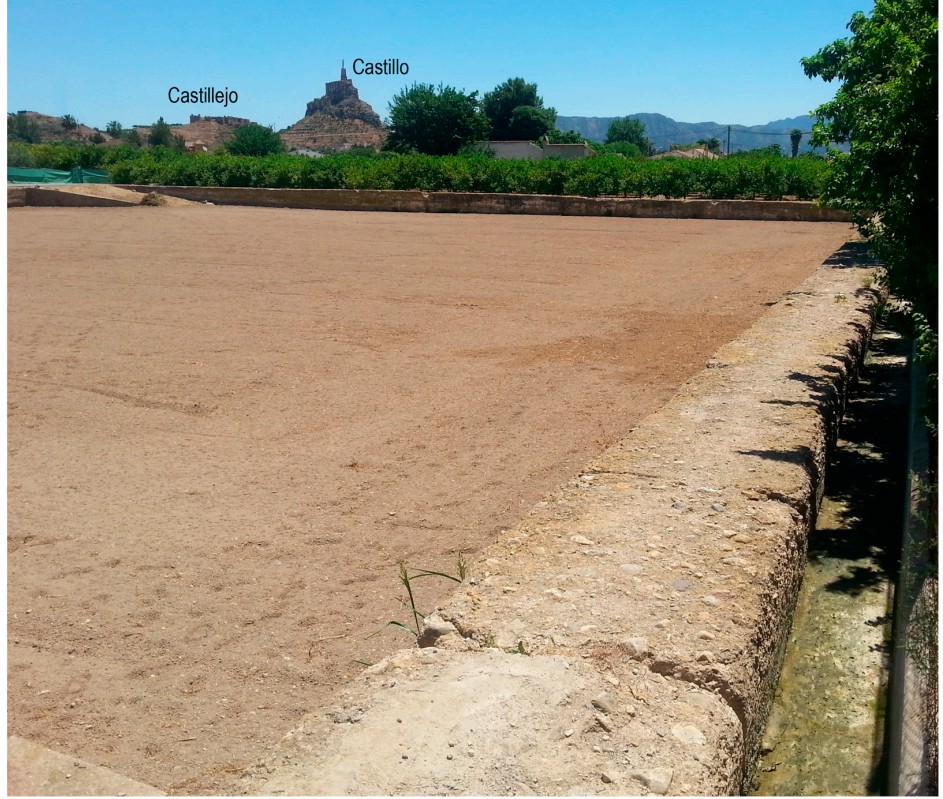

**Figure 6.** Larache. Huerto Hondo. Former pond now used as a cultivation area.

## 2. The Layout of the Estate

Based on the archaeological evidence, the Castillejo de Monteagudo is the most outstanding rural estate on the outskirts of the *huerta* of Murcia at the foot of the two mountain ranges that run parallel to the pre-coastal depression carved by the Guadalentín and Segura rivers.[8] The rivers brought irrigation water to the areas farthest away from the *vega* and to the dryland agricultural areas situated above the *huertas* (Figure 2). Some evidence suggests that these areas belonged to the Murcian state, to the emir or his family, and to various members of the Andalusi aristocracy.

The text of the *Repartimiento de Murcia* records the existence of a series of estates that belonged to the urban elite. They were known as *rahales* or *rafales*, and were located both in the region of Campo de Cartagena, where they seem to have been the most widespread form of land ownership and in the peripheral areas of the Segura-Guadalentín valley. The estates mentioned in the written sources seem to correspond with those that have been identified archaeologically. These estates are easy to distinguish from peasant villages, which are generally located in the middle of the *huerta*, and from other estates called *reales*, also of an aristocratic nature, most of which were situated near the city of Murcia (Jiménez Castillo 2018a, 2018b).

As such, two types of property coexisted in the hinterland of the Andalusi city, peasant-run properties and large estates, which could give rise to disputes, according to Ibn 'Abdūn's treatise:

> "The cadi should order that the people of the villages (*qurà*) ensure that every village (*qarya*) has a guardian (*ḥāriz*) to prevent their properties (*amwāl al-nās*) being treated as if they were free/abandoned (*sā'iba*), because the peasants (*ra'iyya*) look upon the properties (*māl*) of the people of the city (*ahl al-ḥāḍira*) as being lawfully (*ḥalāl*) theirs".[9]

We know of some of these properties from the few surviving remains as well as from occasional mentions in the written sources, which, in general, are lacking in precise topographical information. For instance, the biography of the *ulema* Abū Bakr al-Ansārī claims that he was offered the position of *cadi* during Ibn Mardanīš's reign *"but he refused and preferred to isolate himself and remain in his rustic property on the outskirts of Murcia…"*, until, towards the end of his life, he agreed to take pupils and pass on his wisdom (Carmona 2000, p. 100).

These properties were located at the foothills in association with a wadi from which rainwater could be diverted to irrigate dry crops (Figure 7). Another shared characteristic was the presence of a residence built on a high place, from which the agricultural estate could be monitored while displaying the owner's authority across the landscape. The following paragraphs describe a few examples of this type of property.

One is located in the village of Cabezo de Torres, about 450 m west of the castle of Larache (Figure 3). The remains of the fortified residence are situated on the top of Cabezo de Abajo (Navarro Palazón and Jiménez Castillo 1993, p. 450; Manzano Martínez 2007, pp. 267–69). They comprise a medieval quadrangular building (30 × 30 m) and four towers in the corners, running flush with the wall. On the northern side, an outer wall forms a narrow passageway with the main wall. No information exists about the interior of the enclosure because it is filled with earth and debris. The remains of a free-standing quadrangular tower, the function of which is uncertain, are preserved 20 m to the north. The remains of a large reservoir, which seems to have been used to irrigate the estate, were identified about 250 m to the north, at the foot of the hill and beneath a gully that runs between the *cabezo* and a neighbouring hill. The estate must have been water-fed by the Rambla del Carmen, and a diversion dam, the level of which was raised several times, has been identified upstream,[10] although no remains from the Andalusi period have been found to date. A will from 1883 records the existence of a dryland farm, known as the Carmen, on the Girona headland, near the edge of the ravine and above the Churra la Nueva irrigation channel.

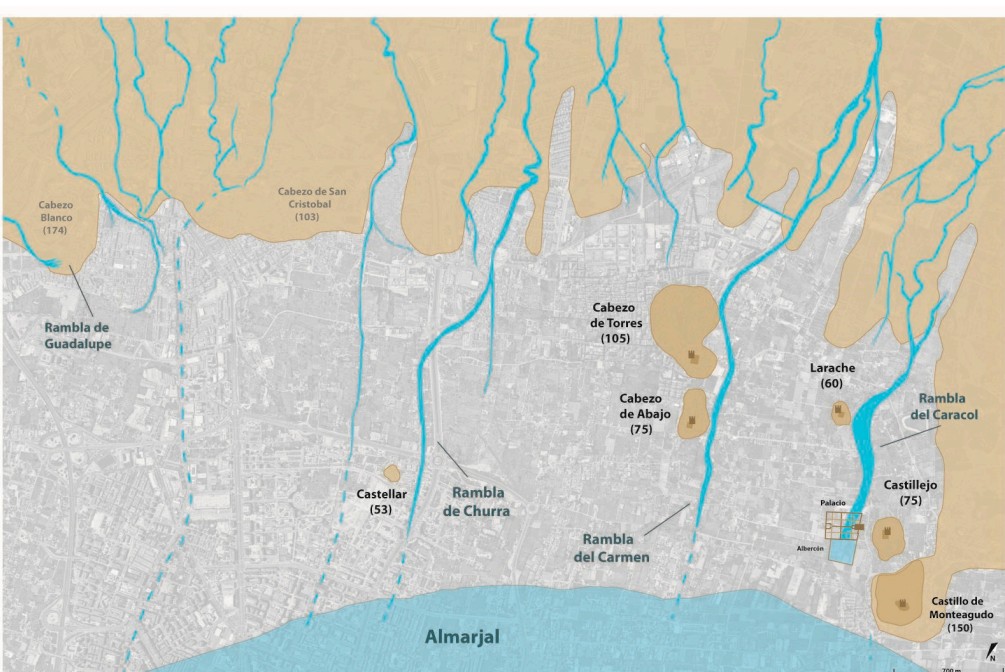

**Figure 7.** Orography and wadis in the Monteagudo-Cabezo de Torres sector. Image by Almela Legorburu (2015).

Another estate must have existed a little farther west, in association with the Churra wadi, where two large fragments of an Andalusi dam were found washed away by floods from their original foundations, which are nearby.[11] The presence of these irrigation infrastructures is indirect proof that a rain-fed farm existed there at the time. The dam is located next to the former estate of Lo de Casas, which was recently demolished. In a testamentary deed of 1883, the estate and its irrigation system are described in a manner that matches the original medieval layout:

> *In the centre of this farm there is the farmer's house [...] It also has a wine cellar, stables, and a barn; an oil mill [...] and a cistern for the supply of drinking water. [It holds] the right over irrigation water from the Rambla de Churra [...] for its exclusive use, for which purpose the waters are distributed by means of various masonry works...* (Jiménez Lacárcel 2019).

It is likely that there was another estate, including a house, on the Castellar del Churra knoll, where the ruins of a traditional dwelling survive today (Figure 7). The fragmentary remains of rammed earth walls could be medieval. This is a privileged location, on an isolated promontory in the middle of a very rich agricultural area.[12]

On the other side of the valley, the southern ranges of Carrascoy and Cresta del Gallo are rich in water springs whose flow, unlike that in the hills that close off the *vega* to the north, has remained plentiful until recently. Several fortifications and other archaeological remains found on the foothills suggest that there were several farmsteads in this area. Despite having ready access to spring water, these estates also used the water from the nearby *wadis* (Figure 2).

From east to west, the first evidence is found in Algezares and in the area around the sanctuary of La Fuensanta. At the Cabezo de Algezares, where a chapel stands today, substantial structural remains, specifically concrete walls, which may correspond to a fortified hilltop residence that fully fits the model described, have been attested. It is possible that the Andalusi structures located some 100 m to the SW of the sanctuary of La Fuensanta, including the remains of a small quadrangular tower (the walls are 6.25 m long and 0.40 m thick), are related to this hypothetical estate (Manzano Martínez 1997). The tower has an entranceway (0.65 m wide) at ground level in the western corner, and the

interior walls preserve remains of red plaster. Approximately 200 m further down, there is a rectangular construction (8 × 13 m), which could be domestic in nature. It is enclosed by rammed earth walls, 0.55 m to 0.60 m thick and preserved to a height of between 0.80 m and 0.84 m. Several hydraulic structures are visible around these buildings, specifically settling basins associated with a spring, a section of channel, and a cistern. It should be noted that archaeological materials are completely absent, so its attribution to the Islamic period, based solely on construction techniques, must remain tentative. According to José Antonio Manzano, the former tower could be identified with one of the two late medieval written sources located in this area: La Fuensanta and El Sordo (Manzano Martínez 1997, p. 443). The documents, however, could refer to something else, as the Rambla del Sordo is located further west, next to the sanctuary of La Luz, and a 15th-century document mentions that the spring located next to the convent of Santa Catalina was also called "Fuen Santa" (Nieto Fernández 1996, pp. 475–76).

The area where the convent now stands is praised by Andalusi sources, which refer to it as Ayelo. According to Pocklington (1987, p. 196), this evolved into the current toponym Verdolay. Abū l-FidĀ' (1848, p. 256) reports that,

> "Murcia [. . .] has a [good] number of places of leisure (*muntazahāt*), among which are al-Rašāqa (the Arrixaca district), al-Zanaqāt, and the Ayelo mountain, below which there are orchards (*basātīn*) and a plain (*basṭ*) into which the water from the springs ('*uyūn*) pours".

Al-Ḍabbī writes that Ayelo was a "place situated to the south of Murcia", where there was a *rawda* (cemetery) in which illustrious figures were buried (Pocklington 1987, p. 195). The entry dedicated to the monastery of Santa Catalina in the archaeological charter of the Region of Murcia points to "the existence of an Islamic necropolis in the southern half of the sector, on Ibero-Roman levels" (Jiménez Castillo 2013, p. 338). One of the verses of Ḥāzim al-Qarṭājannī's *Qaṣīda Maqṣūra* reads, "And beauty made a halt in one of the residences of the sierra (*magānī jabaliyyāt*), the most exalted (*ajalla*) of which is Ayelo (*Iyāla*)" (Pocklington 1987, p. 195). The commentator of the *Qaṣīda*, al-Šharīf al-Ġharnāṭī, states that "Ayelo is a place in Murcia, and I have been told that it was called *Fadlakat al-Uns* (All the Joys), because all kinds of amusements could be found there".

It is very likely that the Andalusi castle of Santa Catalina, located on a hill in Verdolay, was intended to protect the aristocratic estates of Ayelo and that it even worked as a fortified residence itself. At the base of the castle, there is a relatively flat area, cut by two ravines, called *Monteliso*, in which the Franciscan convent of Santa Catalina currently stands. In the convent's garden, watered by a nearby spring, the remains of a bath complex and of an Andalusi aristocratic residence have been excavated (Jiménez Castillo 2013, pp. 337–42). This complex must have been supplied by the spring, which also fed a pool situated at a distance of approximately 800 m in the SE corner of the modern estate *Estación Sericícola* (González Simancas 1997, vol. I, pp. 439–40; Manzano Martínez 1997, p. 451). The pond is quadrangular in plan (9 × 13 m, interior dimensions), with 80 cm thick mortar walls; other formwork walls can be seen to the south of the basin, but it is not possible to know their function.

The town of La Alberca is in the westernmost sector of the site, where Robert Pocklington (2017) has recently located an estate that belonged to the Ibn Arabī family of the Ṭayyi' clan. According to al-Maqqarī, "the residence of the Ṭayyi' (Manzil Ṭayyi') in al-Andalus was in Murcia" and, in fact, the first Castilian settlers during the 13th–15th centuries referred to a place located at the end of the Salabosque road and, therefore, in the modern town of La Alberca, as Mezlatay (Figure 2). Like the previous estates, it was located on the foothills and may have benefited from sporadic water flows running down the nearby ravine or from the many springs that traditionally existed on this hillside around the Valle Perdido. The place name La Alberca, which has been officially attested since the 17th century, refers to the reservoir in which irrigation water was stored and even to the village of Mezlatay. The importance of this place is attested by the fact that Alfonso X, after

granting it to the city of Cartagena, demanded it back and awarded it to Lorenzo Aben Hud, son of King Ibn Hūd (Torres Fontes 1960b).

The remains of the two buildings of the Portazgo are located at the foot of the Puerto de la Cadena, which connects the fertile plain of Murcia with Campo de Cartagena. They are approximately 800 m below the unfinished castle of La Asomada. The two structures were built very close to one another, a few dozen metres above the Rambla del Puerto, next to the old Cartagena-Toledo road, which is mentioned by al-ʿUḏrī, al-Bakrī, al-Zuhrī, al-Idrīsī, and al-Ḥimyarī, among others. These buildings present the same construction technique, with extremely regular courses, as the castle of La Asomada, and they were also left unfinished. It is argued that La Asomada is a Mardanisid building based on its architectural characteristics (these buildings are not mentioned in the written record), especially the peculiar layout of the towers, which form incoming corners similar to those found in the buildings of Monteagudo; for this reason, we believe that the Portazgo structures are also Mardanisid. A series of hydraulic structures were built in the wadi that runs between La Asomada and the Portazgo (they are probably associated with the latter) to collect water from a spring, the wadi, or both (Manzano Martínez 1997, p. 466; 2002, p. 667). Today, these structural remains overlook an area of small traditional orchards and an early 20th-century villa, which suggests the continued use of the area for both production and leisure. It is interesting to note that the special role played by the Castillejo estate on the northern edge of the *huerta* was mirrored by this monumental complex of La Asomada and El Portazgo, which the evidence suggests is the Alhorra *rahal* of the late medieval Castilian documents (Torres Fontes 1960a, pp. 225–34).

## 3. Hydraulics

These medieval estates cannot be analysed solely on the basis of their architectural remains, however monumental they may be, as they were primarily viewed as productive investments, a source of both wealth and prestige. In this sense, it is essential to consider the physical and climatic characteristics of the region and, based on these, the hydraulic infrastructures built to supply both dryland and irrigated crops while meeting the water needs of gardens and residences.

In the semi-arid climate of south-eastern Spain, like in much of the Mediterranean basin and even areas of the Middle East, rainfall regimes are highly irregular, both inter-seasonally and interannually: overall low precipitations can be compounded by summer droughts that can last for up to nine months. However, occasional heavy downpours occur, leading to sudden runoffs in ravines, gullies, wadis, etc. (Gil Meseguer et al. 2015).

### 3.1. The Marshlands

The middle valley of the Segura is a flat or gently-sloping area, poorly drained as a result of the river's meandering course; numerous wadis and even a tributary river, the Guadalentín, which formed an inland delta to the south, drained into the river (Figure 2). The area abounded in endorheic marshland and swamps, which have been gradually drained to reclaim land for agriculture, through the excavation of drainage channels or *azarbes* to lead stagnant water (and later surplus irrigation water) into the river, followed by irrigation channels.[13]

The Arabic word *al-Marǧ* (marshland) features for the first time in verse 295 of the *Qaṣīda Maqṣūra*: "*And how many a pleasant walk from worldly vice we took in Muntaqūd and al-Marj!*". According to Pocklington (2016, p. 1030), it means 'meadow, pasture', but it evolved to mean 'marshy place', so that the word is at the roots of the Spanish term 'almarjal'. The *Repartimiento* mentions these wetland areas in the Murcian Plain on several occasions, which suggests that different areas of marshland existed on the north bank of the river, at least in Aljada, near the Torre de las Lavanderas and in Monteagudo.

With regard to the marshland in Aljada (Figure 2) (a *pago* that approximately coincides with the modern districts of Puente Tocinos and Llano de Brujas), the third and fourth land distributions (dated to 1266) feature several land grants totalling 20 *tahúllas*. Most

of these entries do not include the value of the *almarjales*, as this is added to that of the associated irrigated land, except in the following entry, in which the marshland is value-assessed separately:

> "In Aliada Pero Thomas has vi *taffullas*, which are i *alffaba* and vi *oehauas*. He has, in what belonged to Guillem de Ualuerde and Guillem dez Broyll, vii *taffullas*, which are iii *alffabas* and a quarter. He has v *taffullas damarjal* by Enadença. Total xvii *taffullas*, which are v *alffabas*" ([Torres Fontes 1960a](), p. 35).

Pero Thomas, therefore, was given 13 *tahúllas* of irrigated land valued in 5 *alfabas*, together with 5 *tahúllas* of marshland, a total of 17 *tahúllas* [sic] "which are V *alffabas*"; in other words, the marshland did not add any fiscal value to the grant.

In the fifth distribution, which took place between 1272 and 1273, a marshland located near the famous Torre of the Lavanderas (Figure [2]()) is also mentioned, "...they gave Benito Ferrem x *ataffullas* in the *amarial* of the tower of the lauanderas" ([Torres Fontes 1960a](), p. 234).

In another passage of the fifth distribution, a marshland area was granted to Guillem de Narbona near Monteagud, between the Mayor and Monteagudo *azarbes*, "an *almarjal* left over from all the partitions, which is between the lands of Godiaçibit and the drainage canal of Montagut and the Reyna estate, as far as the Mayor drainage canal, ccc *ataffullas* if possible, except what we gave to Andreu d'Orrit x *ataffullas*, and to Domingo Pintor x *ataffullas*, and to Domingo Perez, nephew of the bishop, x *ataffullas* and to [...]" ([Torres Fontes 1960a](), pp. 238–39). This fragment of the *Repartimiento* suggests that the size of these marshes was, at this time, approximately 340 *tahúllas* (the 300 granted to Guillem de Narbona plus the 40 awarded to the other settlers). Relevantly, the text also mentions two drainage channels in the sector (*azarbes* Mayor and Monteagudo), which leads us to believe that they were dug during the Andalusi period.[14] The text, therefore, yields information about three important aspects of the marshlands of Monteagudo: first, they could be owned and exploited; second, owing to the topographical references in the text, they can be situated between the two *azarbes*, the Real de la Reina and the area of Cudiaçibit; third, around the time of the conquest, they had already been partially drained and put under cultivation, although other parts were still in its natural state.

During periods of demographic and agrarian downturn, the partial or total abandonment of the drainage network contributed to the natural regeneration of the wetlands, a recurrent phenomenon from the third quarter of the 13th century onwards. This seems to be indicated by an Alphonsine privilege dated 18 May 1267, which gave the city council the power to appoint two good men per parish every year "...to keep the major irrigation channels of the *huerta* clean, to stop the formation of marshland...". In 1308, King Ferdinand IV granted Juan de la Peraleja 1000 *tahúllas* in the marshes of Monteagudo, a much larger property than that granted to several individuals in 1273, which together only amounted to a third of that size ([Torres Fontes 1980](), p. 88). If both documents refer to the same thing, they strongly suggest exponential growth in the size of this wetland. A description of the expansion of these marshlands and their use as a hunting ground can be found in Don Juan Manuel's *Libro de la Caza* (1325):

> "There are many herons and *vitores* in the marshland of Monteagudo, but they are very difficult to kill with hawks, but, on the banks of the marshes, ducks can sometimes be found in places where they can be hunted with hawks [...] Sometimes they are also found in the lagoon at the Menoretas gate" ([Blecua 1982](), p. 580).

However, the expansion of wetlands became a health hazard, meant the loss of agricultural land, and posed a threat to crops in the surrounding areas, especially trees. In order to remedy this situation, a 1332 decree by Alfonso XI reiterated the need to clean the "two major channels where the waters run off so that the estates are not lost to excess water" ([Torres Fontes 1975](), p. 22). However, it does not seem that this measure was effective, as the records indicate that the marshes continued to be sown with water-demanding crops,

such as rice and flax, the expansion of which is well attested during the reign of Peter I (Martínez Martínez 2013, p. 49).

In 1408, the regent Don Fernando authorised the council of Murcia to drain, subdivide, and distribute the marshes of Monteagudo, for which the *azarbe* needed to be repaired (Martínez Carrillo 1997, pp. 151–59); the operation was repeated in 1490 "so that the lands and estates may be sown with wheat" (A.M.M.U., Leg. 4272, n. 82). In the late 16th century, the eastern sector of the marshland, an area known as La Udienca, between El Esparragal, La Cueva, and Santomera, was drained by the Hieronymite monks of San Pedro de La Ñora (Canales Martínez and Ponce Sánchez 2021, p. 31).

*3.2. Wadis (Ramblas)*

One of the most characteristic physical features of the northern face of the pre-coastal depression between the districts of Guadalupe and Monteagudo is the wadis (*ramblas*). These watercourses, which run N–S (Figure 7), collect and channel copious amounts of rainwater into the alluvial plain, causing considerable erosion. These watercourses run dry most of the year but often experience sudden flows due to torrential rains, generally during the autumn and the spring. This water carries large quantities of sediments ("turbiones").[15] The wadis tend to enter the Segura floodplain perpendicularly, flowing into the river or the low wetlands.

Works to mitigate the traditional threat posed to the hydraulic and productive macro-system of the *huerta* of Murcia by overflowing wadis are attested from an early date. One way of doing this was to divert their episodic flows towards dry farming areas alongside their banks by means of dams, sluice gates, and channel networks. Early evidence from the estate of Monteagudo includes the creation of a cross-shaped garden over the silted-up bed of a former wadi in the mid-12th century, which suggests that earlier anti-flood measures had been successful; we shall deal with this palatial complex in detail below.

Four major ravines, known as Cabezo Blanco, Churra, Carmen, and Caracol, along with smaller ones, e.g., Figueretas (Molina Molina 1989, no 236), run through the area under analysis.

Cabezo Blanco, also known as "la del Zoco" or "de Echevarría", is the westernmost *rambla*. It is located between the hamlets of Guadalupe and Espinardo (Figure 8). The head of the *rambla* (174 m above sea level) is at a distance of 1800 m from the dam (74 m above sea level) (i.e., a difference in altitude of 100 m; over 5% gradient). The diversion dam, known as Azud de Guadalupe,[16] is 15 m long and presents an arched structure, with an opening, 1.70 m wide and 1.00 m high, at each end. An initial examination of the remains ruled out a medieval date, which does not necessarily mean that an Andalusi dam did not exist there or nearby.

The Rambla de Churra is located 2.6 km to the east of Cabezo Blanco (Figure 9). Although it does not flow directly into the Castillejo estate, its Andalusi dam is still a valuable illustration of the area's typical flood-control systems. As in the case of the Rambla del Carmen, its course can only be made out in an aerial photograph taken in 1928, which shows that, upon entering the orchards, the wadi was channelled alongside the Churra road towards Murcia, losing its natural morphology.

The Rambla del Carmen also enters the plain through the modern Cabezo de Torres district from the north (Figure 10). As noted, a diversion dam was found upstream. The *rambla* is heavily altered, turning into a path through the *huerta*, but its course can be clearly distinguished in the 1928 photograph: a first meander tangentially touches the rocky outcrop to the north of Cabezo de la Cruz, followed by a second meander that runs separately around Cabezo Collado; then the wadi turns, skirting Cabezo de Abajo before running straight 550 m to the south, where the riverbed becomes indistinguishable as it meets the medieval irrigation channel of Zaraíche perpendicularly. Overflooding in this wadi posed a substantial threat to cultivated areas and even irrigation channels, as reflected in a council agreement dated to 1445, which aimed to prevent the effects of the autumn floods by building a flood wall (*atochada*).[17] The lower course of the wadi, already within

the *huerta*, is outlined by lime concrete walls, which may explain why it is referred to as a "canal" in 15th-century records (*rambla* or canal of Oriaque).[18]

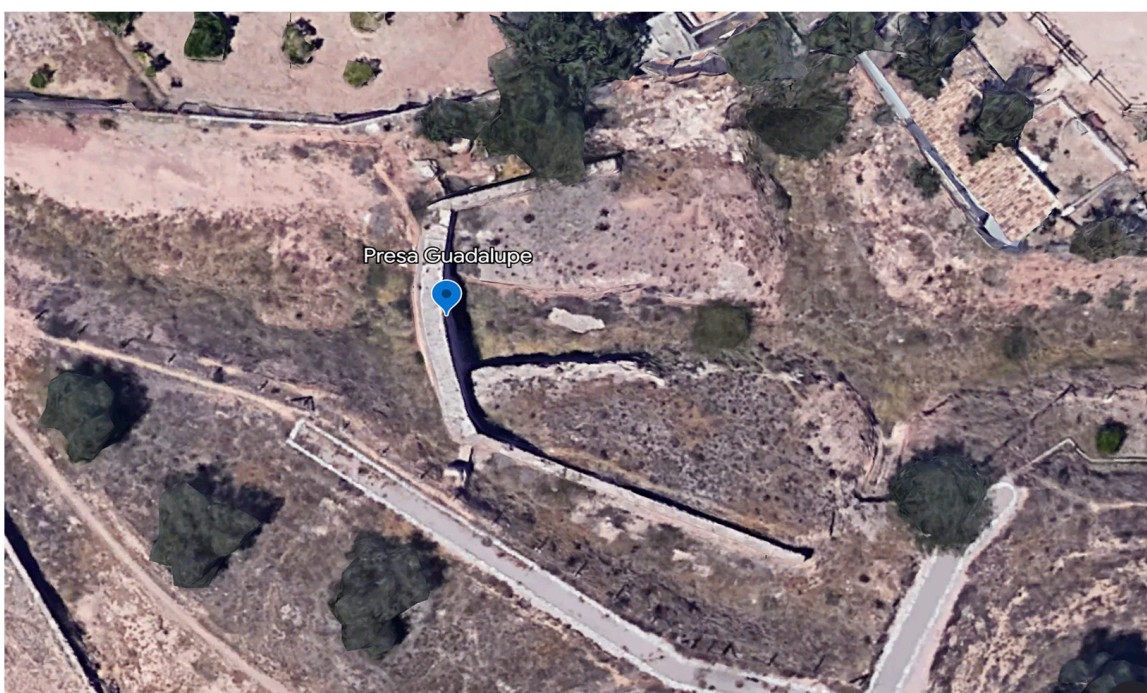

**Figure 8.** Guadalupe wadi and dam. Source: Google Earth.

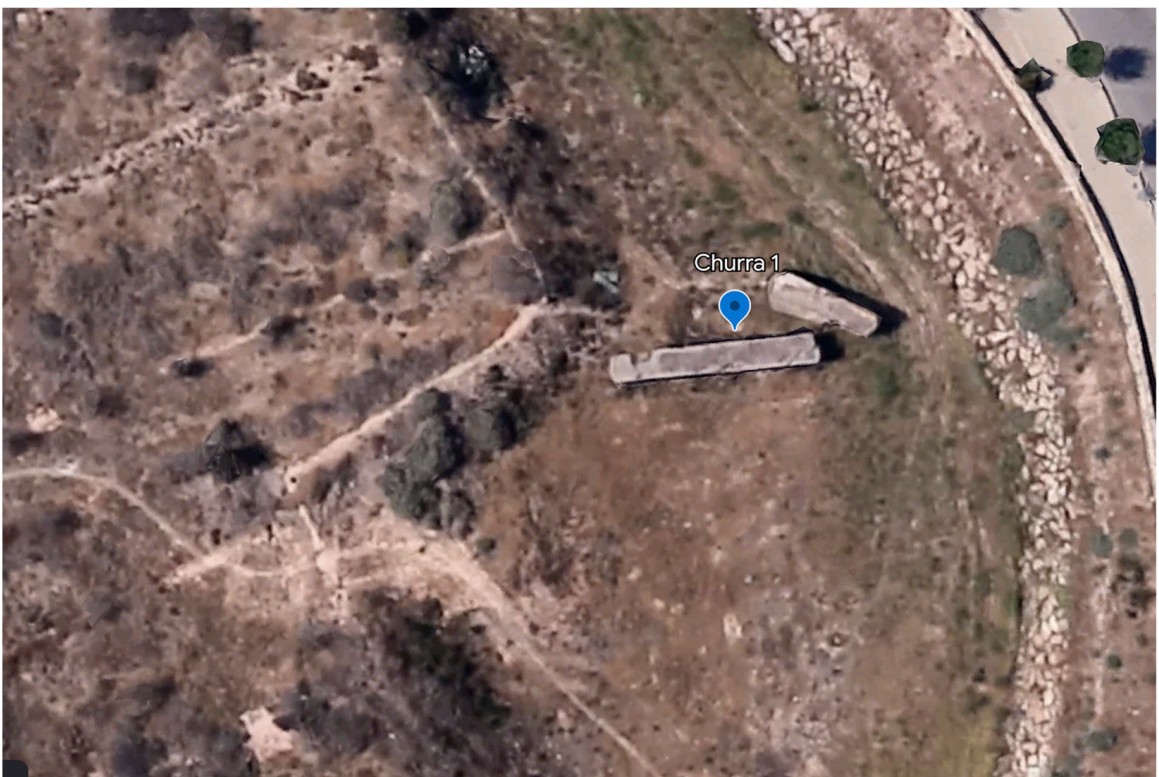

**Figure 9.** Churra wadi and dam. Source: Google Earth.

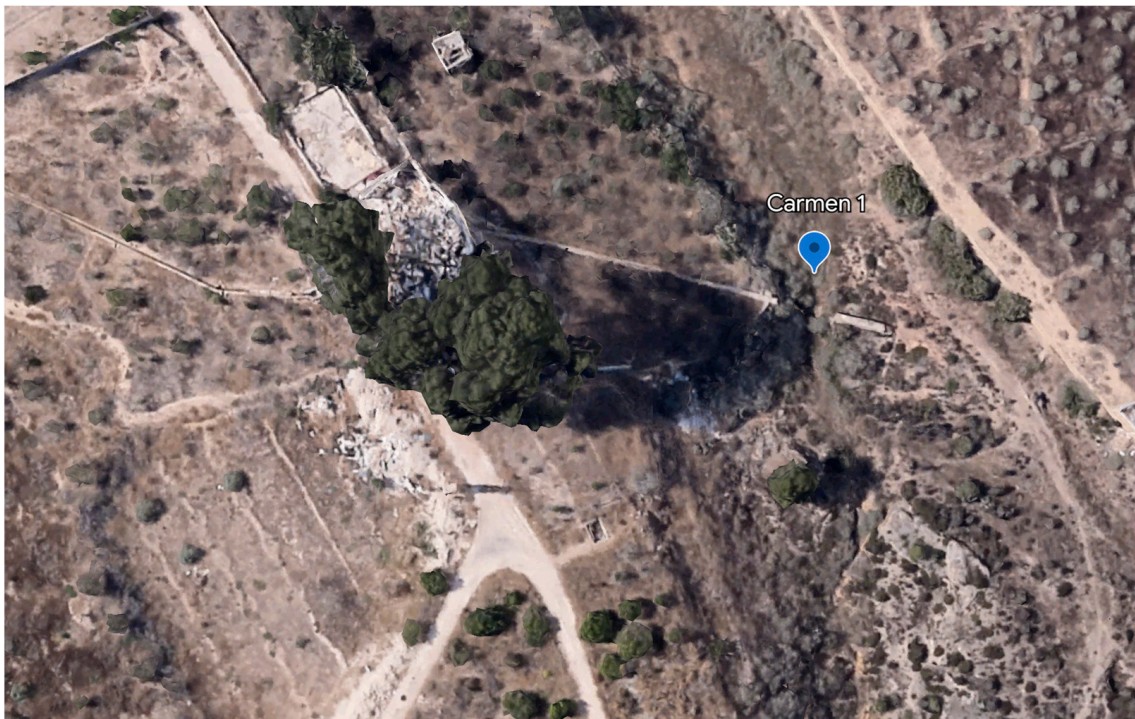

**Figure 10.** El Carmen wadi and dam. Source: Google Earth.

The Rambla del Caracol is the easternmost wadi and the one that affects the estate of Monteagudo most directly, as it flows into the palatine area (Figure 11). In 15th-century records, it is referred to indistinctly as the Rambla del Caracol or Alabrache; the former name seems to allude to the fact that it dies near the homonymous irrigation channel.[19] Its final section is depicted in a plan of the Murcian *huerta* by Pablo de Villar, dated to 1809, in which the tail-end of the wadi can be seen to form a delta. The map matches 15th-century descriptions, which proves that by that date, the wadi no longer ran across the irrigation channel. Although no longer in place, the old course is visible in the shape of cultivated plots; it is clearly discernible in the fossilised section at the foot of the hills of Larache (east) and Castillejo (west); these traces disappear 400 m to the south, at the foot of the Monteagudo Castle, where the wadi flowed into the marshes. It is interesting to note that the large cross-shaped garden was built upon the old bed of the wadi in the 12th century, which shows that the watercourse was already being diverted to irrigate the fields upstream.

No evidence of the dam assumed to have existed in the Caracol wadi has been found to date. However, it is thought that the rainwater control system used in this wadi was identical to that used in the others, namely a flood diversion dam (of concentrated runoff) in the bed of the wadi and terraces in the cultivated fields below. The dams cut across the bed of the wadi or ravine, diverting the water towards one or two ditches located at the head of the channel system. The diverted water was distributed through a system of canals that branched out to form extensive networks across the terraces. Their extension and complexity depended on the local orography and the size of the cultivated plots. The surplus flow was returned to the wadi through drainage channels. This system, which for centuries has provided additional irrigation water to rainfed crops, also contributed to minimising the damage caused by flooding, especially in the lower sections of the system.

These systems are behind characteristic landscapes known as "irrigated peatlands", "assisted dryland", and "improved dryland", which are typical of south-eastern Spain (Gil Meseguer et al. 2015; Hernández Hernández et al. 2019). In the book of the *Repartimiento*, these crops are referred to as "*alfait*". Asín Palacios (1940, p. 58) linked the Valencian toponym *Al-fait* to the Arabic *al-fayd*, which means "flood, inundation, overflow". Afterwards,

Torres Fontes (1971, p. 37) and Pocklington (1984, pp. 281–84) examined medieval Murcian texts and suggested that Andalusi *alfait* irrigation systems existed in Tiñosa, Sangonera, and in some "*rahales*" in Campo de Cartagena. The *alfait* was, therefore, one of the ways to irrigate *alvar* land, together with waterwheels (Torres Fontes 1959, pp. 76–77). The *wadis* must have inspired the construction of the earliest irrigation systems in the region.

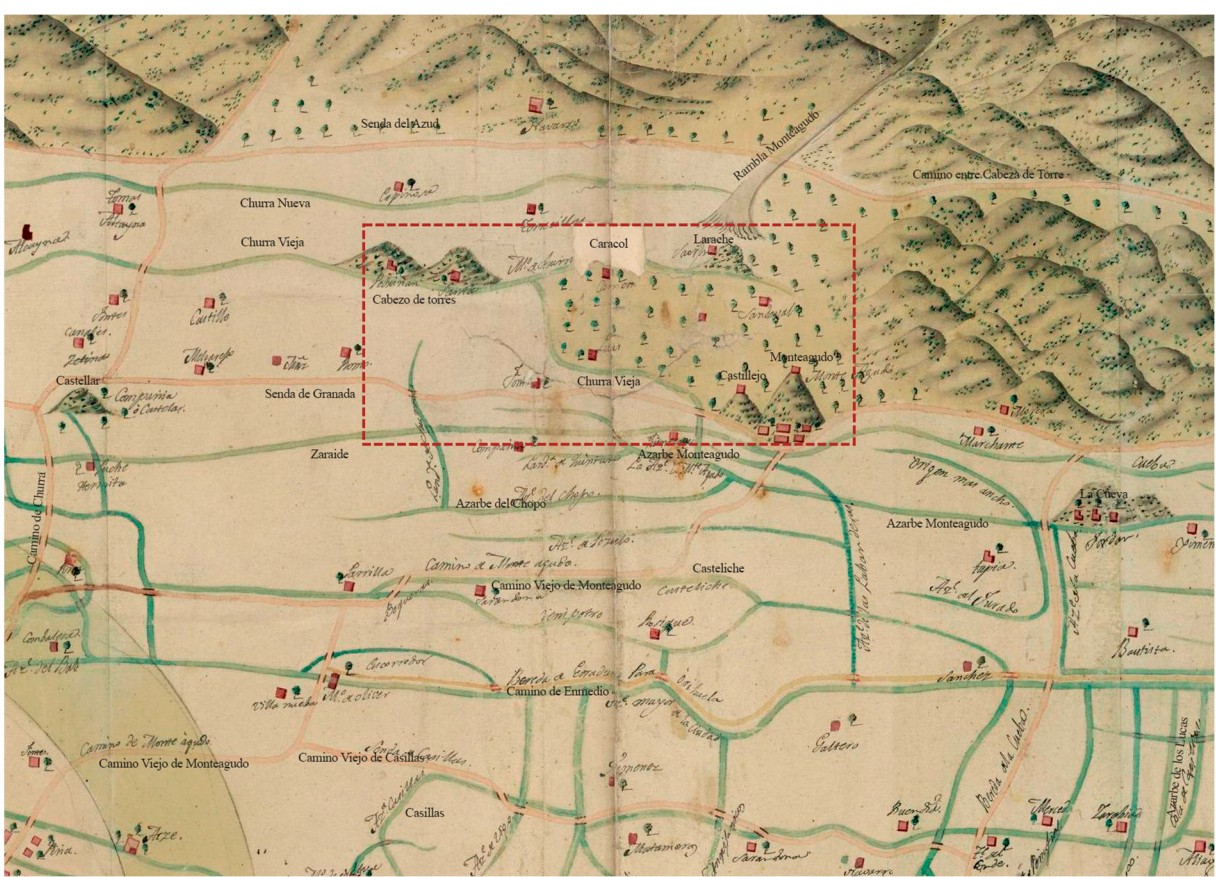

**Figure 11.** The hydraulic system of the Monteagudo-Cabezo de Torres sector, as well as the Larache or Caracol wadi, in the map of Pablo de Villar (1809).

*3.3. Cisterns*

In addition to irrigation channels, large water cisterns have also been identified in the study area (Figure 3). Except for the one found in the palace complex on the plain, which will be discussed later, none have been archaeologically examined, so their construction and abandonment dates are uncertain; however, it is thought that all of them were built in the Andalusi period to serve the needs of the nearby country estates.

One is located approximately 250 m north of the Andalusi remains at the top of Cabezo de Abajo, in the area of Cabezo de Torres, close to a gully that runs between the knoll and a nearby hill. Rectangular in shape, the cistern is 90 × 78 m in size, and its mortar walls are 2.30 m thick. We cannot rule out the possibility that it was supplied by the Rambla del Carmen, which still flows in a N–S direction close to its eastern side. The Churra la Vieja irrigation *canal* (*acequia*) runs from the west into the NE corner of the cistern, whose walls form a channel, so it can be deduced that the *acequia* is later than the cistern. The aerial photograph reveals that the plots near the hill and southwards, as far as the Senda de Granada, some 100 m on either side of the Rambla del Carmen, are peculiarly arranged. Unlike the agricultural plots, the layout of the terraces does not exactly follow the contour lines, which could indicate the area of land served by the reservoir before the construction of the Churra la Vieja irrigation channel.

The old quadrangular cistern known as "Huerto Hondo" is situated to the west of the Larache Castle. It is 60 m × 58 m in size, with 1.50 m thick walls; the walls are 1.50 m high on the outside but lower on the inside because the cistern is partially filled with cultivated soil (Figure 3). The presence of red plaster on the inside, the existence of spillways, and accounts by local farmers, who report that the cistern has a now-buried paved floor, confirm that it was a cistern used to store irrigation water, and also probably for recreational activities. Owing to its proximity to Larache, it has always been assumed that it is related to this building. However, since Larache is probably later than the Castillejo palace, we cannot rule out the possibility that it was originally used to irrigate the Castillejo estate because it is on higher ground than the Castillejo reservoir, which is set too low for this purpose.

To the south of the Huerto Hondo cistern, the much larger Monteagudo reservoir is situated below the Castillejo palace, to the southwest (Figures 12–14). Judging by the extant remains of three of its sides, which are still visible in the modern orchards, it must have measured 161 × 136 m; no remains of the northern side exist, and it very likely ran under the Granada road.[20] The eastern wall is particularly well preserved; it survives to a maximum thickness of 2.40 m, although some sections of this wall are heavily deteriorated or are missing. The Zaraíche irrigation channel runs W–E into the southwestern corner of the reservoir, where it abruptly turns SW to skirt the spur of the Monteagudo castle. The land irrigated by this *acequia* roughly corresponds to that served by the cistern, so it can be assumed that in the mid-12th century, when the cistern was built, there was no such channel. This seems to be confirmed by the fact that the section of the irrigation channel that runs between the pond and the village of Monteagudo cuts across the field system. This area was originally a wetland, drained by the Monteagudo *azarbe*, which strongly suggests that the drainage channel is contemporary with the reservoir.

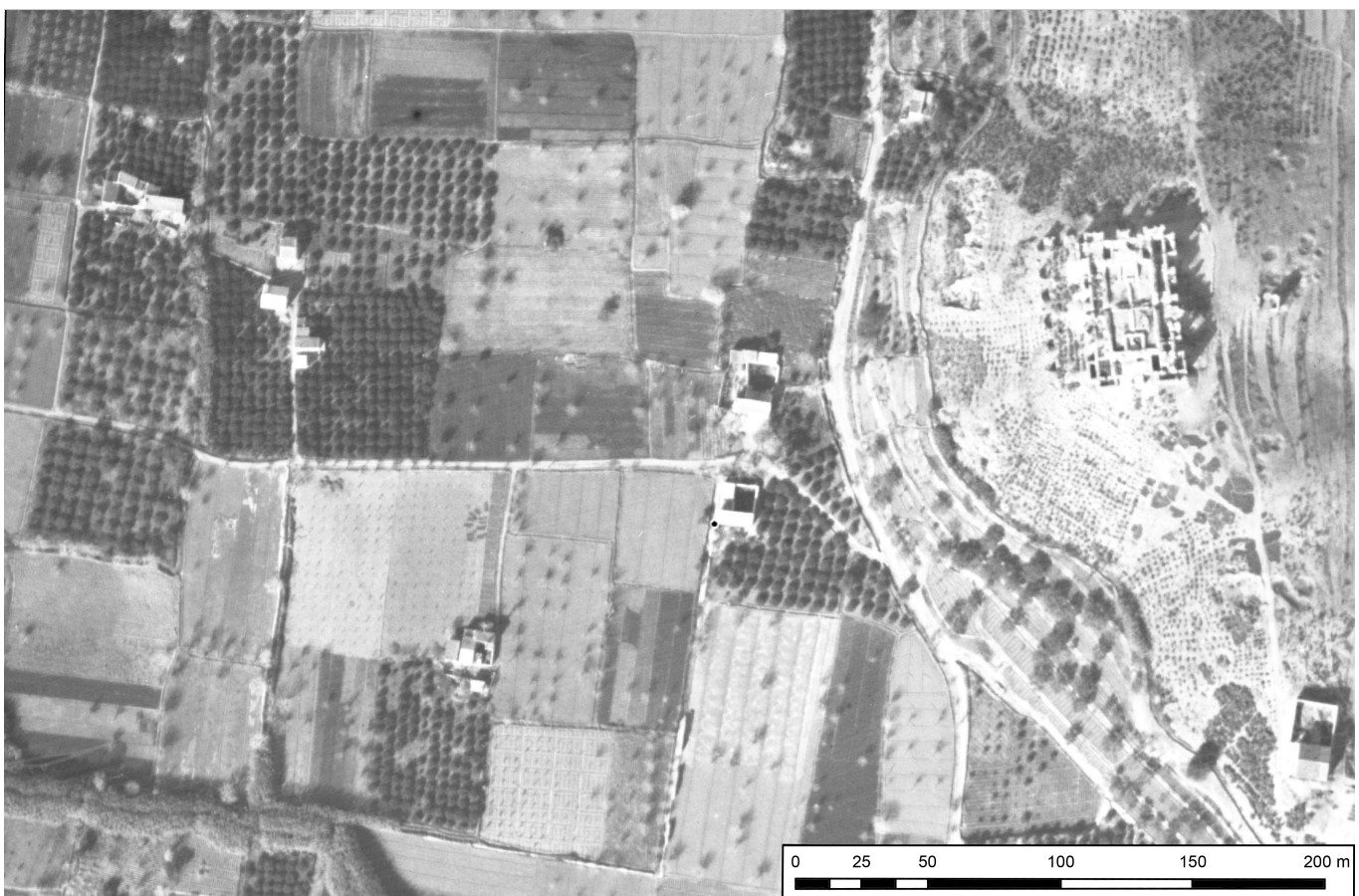

**Figure 12.** Aerial photograph of the Castillejo and the sector of the *huerta* occupied by the palace on the plain. Flight by Ruiz de Alda (1929–1930).

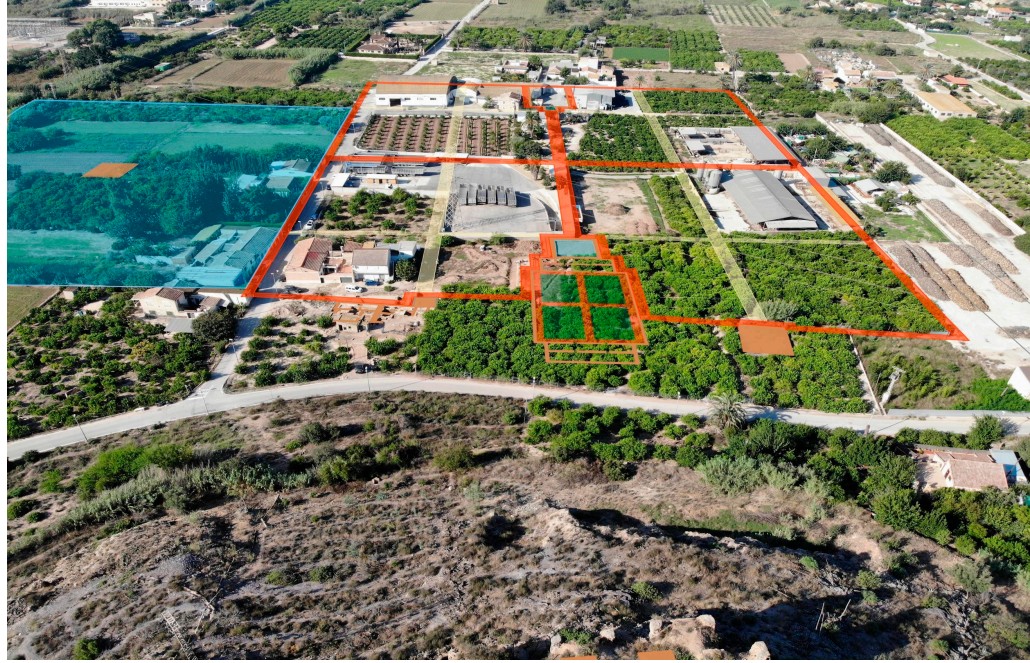

**Figure 13.** Aerial photograph of the Castillejo and the sector of the adjacent orchard. Highlighted, hypothetical restitution of the ground plan of the palace. Flight by Ruiz de Alda (1929–1930).

**Figure 14.** View of the palace from the Castillejo. Highlighted, the preserved structures and the hypothetical restitution of the palace.

The fragment of the *Qasīda Maqṣūra* that refers to Monteagudo and the palace of Ibn Mardanīš mentions an *al-ṣuhayrīj*, referring to a nearby cistern. The word lies at the root of the name of the Zaraíche irrigation channel, which runs directly into the southwestern corner of the Castillejo reservoir. The orchard that now grows inside the reservoir is irrigated by this channel, which also supplies the area drained by the Monteagudo irrigation channel. The latter is mentioned in the *Repartimiento*, so it seems logical to assume that both channels—*acequia* and *azarbe*—were built at the same time, before the Christian conquest. Late medieval documents mention the "Monteagudo pond" near the castle, perhaps the same one mentioned by al-Qarṭājannī. In conclusion, both Muslim and Christian sources dated to between the 13th and 15th centuries prove the existence of a cistern or *zaraíche* in Monteagudo, and this is likely to correspond with the Castillejo cistern just described. The cistern likely lent its name to the irrigation channel (*acequia*), so it can be assumed that the latter is more recent than the former, although both date to the Andalusi period, as suggested by the *acequia*'s Arabic name and the fact that it is mentioned in the *Repartimiento*.

In contrast to the southern mountain range, the mountains to the north of the valley have few springs. Cisterns, therefore, could only be supplied by the irregular *wadis*, diversion dams, or irrigation channels feeding from the Segura River. In principle, cisterns are unnecessary in areas in which irrigation *canals* can guarantee a regular water supply, for instance, the Murcian *huerta*, which is supplied by the river. In fact, no large reservoirs have ever been built in the *vega*, nor is there any trace of them in customary law. These infrastructures are traditionally associated with systems in which water must be managed through dams, as they rely on continuous but scarce water sources, such as springs and *qanats* (underground channels), or intermittent flows, such as *wadis*. The presence in the area of two wadis (Carmen and Caracol) and two cisterns nearby has already been noted. The *rambla* found near the fortress of Larache, a little further downstream from the Caracol's dejection cone, is particularly important here. It is interesting to note that the large cross-shaped garden of the palatine area of the Castillejo was built upon the tail-end of this wadi, putting it at risk of flooding had necessary control measures not been taken upstream. Therefore, it seems logical to assume that when the palatine area of the plain was designed, the torrential flows that sporadically ran down these wadis were already under control and probably, partially stored in these cisterns.

If the Monteagudo and Cabezo de Torres reservoirs were supplied with water from the wadis, it can be assumed that the diversion dams were equipped with sluice gates to divert water not only to the rain-fed crops but also to the reservoirs. A filtering system to stop sediments from entering the cisterns was also probably in place.

Archaeological examples of this sort of water supply system are attested in the steppe regions of modern Syria and Jordan. Water is scarce in the region, with no permanent watercourses, although *wadis* abound. These run dry for most of the year but can accommodate significant water flows when torrential rains occur. Evidence of dams and large rectangular cisterns similar to those associated with the aulic estates of Monteagudo and Cabezo de Torres has been found for both the Roman period and the early years of Muslim expansion. The hydraulic complexes linked to some of the so-called Umayyad desert castles, dated to the early 8th century, such as al-Qasṭal, Qaṣr al-Ḥayr al-Gharbī, and al-Muwaqqar, all of which were part of rich irrigated estates presided by palatial buildings, are particularly interesting. Other desert 'castles' were also associated with remarkable hydraulic systems, including dams and channels, although they lacked large reservoirs (or they have not been located), e.g., Qaṣr al-Ḥayr al-Šarqī, Umm al-Walīd, Quṣayr ʿAmra, and Khirbat al-Mafjar. Rusafa (Syria). The ancient Sergiopolis was largely supplied by runoff water, which involved the construction of an earthen dam~450 m long, to divert flood water into a reservoir. A channel connected with this reservoir carried the water into the cisterns through an opening in the city wall.

Large reservoirs are also known in Aghlabid Kairouan, one of which measures 128 m in diameter. Another one, similar in terms of size and wall thickness to that in Monteagudo (290 × 80.5 × 2.35 m), exists near Tunis (this one was mentioned by Ibn Khaldun). Another

example, near Tlemcen, is 200 × 100 m in size. In Marrakesh, in addition to the large reservoirs of Agdal and Menara, the remains of numerous quadrangular cisterns exist, although in these instances, they were not fed by *wadis* but by *qanats*, known locally as *khattaras*.

Large water reservoirs are also known in al-Andalus, alongside diversion dams, weirs (Spanish *azud* from the Arabic *al-sudd*), and storage dams. A storage dam formed an artificial lake in the vicinity of Segura de la Sierra (Jaén). According to al-Zuhrī, it was built by Ibrāhim ḅ. Hamušk, lord of Segura and Ibn Mardanīš's lieutenant, and some of its remains survive to this day.

The gorge, or *maḍīq,* was closed by Abū Isḥāq ibn Hamušk when he was lord of Segura, with a perfect work of engineering in imitation of Ma'rib's dam in Yemen. He turned the plain into a sea when the water level rose and built no spillways because he wanted the reservoir to overflow through the mountaintops, but the terrain did not help him (Salvatierra and Gómez 2016, p. 309).

If these parallels and their location next to the course of *wadis* are taken into account, it can be argued that the Monteagudo-Cabezo de Torres reservoirs are fossils of irrigation systems designed to divert irregular wadi flows and that they predate the construction of the network of *acequias* (Figure 7). Initially, these *albercones* also had recreational purposes as part of the palatial areas of aristocratic estates. The system based on reservoirs and *wadis* was later superseded by the construction of the *acequias* of Churra la Vieja and its Caracol branch. As pointed out by Manzano, this does not mean that the large reservoirs were immediately abandoned, as it seems that the new channels were used to supply them, creating new potential ways to manage water. This sequence seems plausible in, at least, the Castillejo reservoir, as archaeological surveys and excavations appear to show that the aqueduct to the north of the reservoir was fed by the irrigation channel of Churra la Vieja. Eventually, however, the reservoirs were abandoned, as they were no longer needed for irrigation, and the recreational uses to which they might have been put were probably also extinct, as the royal estates no longer existed.

According to al-Qarṭājannī, the southern sector of the area described was covered in marshlan, and could only be sown after the drainage channels were excavated, a process that began in the Islamic period and lasted well into the Early Modern Age. We know that the Monteagudo *azarbe* existed before the Castilian conquest in the mid-13th century, as it is mentioned in the *Repartimiento*, and it seems logical to think that it may date back to the 12th century, as it drains the area irrigated by the Monteagudo reservoir. Later, this area, including the orchard that eventually grew inside the Castillejo reservoir, came to be irrigated by the Acequia de Zaraíche (Figure 3).

*3.4. Irrigation Channels (Acequias)*

The irrigated agro-ecosystem created during the Middle Ages in the middle valley of the Segura River, known as the *huerta* of Murcia, created conditions for the agricultural exploitation of a large, partially endorheic region whose semi-arid Mediterranean environment offered only limited possibilities for agriculture. Rather than the implementation of a unitary project, this was the result of a slow process that began near the banks of the Segura River and the heads of the main irrigation channels at the Contraparada dam and then gradually sprawled towards the foothills (Figures 2 and 11). Its northern limit has traditionally been defined by the canal of Churra la Nueva, located approximately on the line that separates the plain and the neighbouring mountain range. The canal thus marks the maximum extension of the irrigated *huerta* and the boundary with another radically different space dominated by dryland agriculture and scrubland. To the south of the canal, in the area of Monteagudo and Cabezo de Torres, other channels, which lie closer to the riverbed, represent later phases of expansion of the system; they are, therefore, earlier than Churra la Nueva. The main characteristics of these channels are as follows.

-　　Casteliche-Benipotrox. These irrigation channels are located to the south of the Monteagudo and Chopo *acequias* (Figures 2 and 11). Although they lie outside the study

area, they are important to understand the evolution of the landscape of Monteagudo-Cabezo de Torres, as they must have been the oldest in the area and the ones that marked the northern limit of the *huerta* when the *almunia* of Monteagudo was built. As noted by Pocklington (1990, p. 67), the Mozarabic toponym "Casteliche" does not feature in any document dated to the 13th century, probably because at the time the name applied only to the *acequia*'s first section, which was still in Muslim territory. The Benipotrox irrigation channel is a derivation of Casteliche and joins it again after irrigating some fields near Monteagudo. According to Robert Pocklington (1990, p. 207), the name does not feature in the *Repartimiento* or in medieval *Actas Capitulares*, perhaps because the lands that it irrigates were part of the *donadío* of Monteagudo, initially granted to the queen and later to the church.[21]

- Zaraíche. The *acequia*, 16 km long, emerges from the Aljufía irrigation channel in La Arboleja and runs across several districts before entering the municipality of Orihuela (Figure 2, Figure 3, and Figure 11). The earliest mentions in the written record are in the *Libro del Repartimiento*, which calls it *Açihayrch* and *Açuharich* (Torres Fontes 1960a, p. 230), and in the *Partiçión del Agua* of 1353, which calls it *Çaharrich* (Torres Fontes 1975, p. 56). Pocklington's (1990, pp. 233–35) study of the toponymy of the *Vega de Murcia* points out that all these spellings spring from the same toponym, which refers to a pond or cistern. Pocklington also noted that al-Qartājannī's *Qaṣīda Maqṣūra* mentions a place known as *al-Ṣuhayrīj* ("the little pool"), located in the northern half of the *huerta*. This diminutive form, which was previously noted by Emilio García Gómez (1933, p. 102), could be related to the terms that feature in Castilian documents and to the modern name of the irrigation channel. It irrigated a sector near the Casteliche, Chopo, and Monteagudo irrigation channels to the south. From Cabezo de Torres, downslope to the east, the plots of land irrigated by the Zaraíche *acequia* adopt a fairly standard comb-like layout, with diversion channels running southwards, perpendicular to the main irrigation channel and parallel to each other, flowing into the Monteagudo drainage at the bottom of the slope. A good drainage system was obviously necessary to evacuate the surplus water from Zaraíche; according to the written record, this depression was particularly prone to the accumulation of stagnant water, for which reason it was called *acequia insana de Çaharig* (García Díaz 1989, p. 106).

The land of Zaraíche is divided into long, rectangular properties, perpendicular to the irrigation channel, unlike the plots supplied by Churra la Vieja and Caracol, which are predominantly quadrangular in shape. It would appear, therefore, that the presence of the marshes determined the sequence of colonisation in this sector of the *huerta* in such a way that the Zaraíche irrigation channel postdates Churra la Vieja and Caracol, despite being lower and closer to the river.

- Churra la Vieja. This *acequia* is located to the north of Zaraíche and runs at an altitude of 48 metres above sea level (Figures 3 and 11). It rises from the left of the Aljufía irrigation channel, like Alfatego, a little before the wheel of La Ñora, and crosses the districts of Guadalupe, Espinardo, Churra, and Cabezo de Torres, before dying in the Monteagudo *azarbe*. The toponym *Churra* only appears once in 13th-century documents, when the name *Montanna de Churra* is mentioned in relation to the area of high ground between Murcia and Molina.[22] On the other hand, a 13th-century document mentions a former Muslim landowner called al-Xorri;[23] this *nisba*, which means "the Churra-born", shows that the toponym existed before the Castilian conquest (Pocklington 1990, pp. 213–16). However, the homonymous irrigation channel is not mentioned until the 14th century.

- Caracol. This *acequia* branches out of Churra la Vieja between the two hills of Cabezo de Torres (Figures 3 and 11). From there, the branch runs flat to skirt the northern hill, while Churra la Vieja turns southwards sharply and drops 5 m. After skirting Cabezo de Abajo towards Camino de la Almazara, Caracol takes another sharp turn to cross from west to east more than halfway between Cabezo de Torres and Monteagudo, where it turns southwards again to cut diagonally across the modern fields. The final

section, before it crosses the Zaraíche *acequia*, presents several bends that suggest that its course had to adapt to a pre-existing field system. After crossing Zaraíche, the *acequia* continues in a straight line to flow into the Monteagudo irrigation channel. It is possible that the Caracol branch was the original continuation of the Churra la Vieja *acequia* because after they separate, the latter follows an irregular course and sits awkwardly with the plots that it irrigates. A document from 1453 concerning the property of Alfonso de Cascales suggests that this irregular section between the Rambla de Oriaque (today Rambla del Carmen) and the Larache hill may not be its original layout:

> …between the land they call "of the Pujaltes" and the central path that goes from the canal of the Oriaque stream to the hill they call "Alabrache [Larache]", which used to lead to the old *acequia* of Churra; on the other side, facing the sunset, the olive grove of Pedro Jufre, notary, the central path that goes from the *aluares* (dryland) to the irrigation channel of Churra; from the other side it runs along the road that goes from said canal of Oriaque to the Campillo de Guillen Çeldran; and on the side facing dawn, the hill of Alabrache…." (Martínez Carrillo 1997, p. 75).

Therefore, it is possible that the original Churra la Vieja, after passing between the Cabezo de Torres hills, followed the course of what is now known as the Caracol branch, which runs next to a service road. This would explain several of the 'anomalies' detected in the existing Churra la Vieja canal, such as the sudden drop, the fact that one section sits awkwardly with the surrounding plots, and the odd course of its tail-end, which seems to adapt to a consolidated field-system. If so, these alterations may have been caused by the state of disrepair to the system in the 14th century, which prompted new distributions of water (years 1277, 1329, 1353, 1459) (Torres Fontes 1975, pp. 17–22; Martínez Carrillo 1997, p. 68).

- Churra la Nueva. This *acequia* runs parallel to Churra la Vieja (200–300 m to the north) at an altitude of 51 m above sea level, and, unlike those described above, it is not supplied by the main irrigation channel of Aljufía, but directly by the river (Figures 2, 3 and 11). It marks the traditional northern boundary of the *huerta*, above which extensive olive groves existed. It first features in the written record in the 15th century (Martínez Carrillo 1997, p. 74). By 1504, it had more than sixty *brazales* (subsidiary canals) before reaching the *pago de Alabrache*, but the network did not meet the demands of irrigators: they complained of sizeable losses of water in the top section because the mouths were all at the bottom level, which compromised the irrigation of trees in the lower sections. The situation in Churra la Vieja must have been similar, as both the written record and the 1809 plan report that the *pago de Alabrache* was sown with dryland crops. However, it cannot be said with certainty that these date to the 14th or 15th centuries, as it is perfectly possible that the *acequias* were rendered unusable in the late 13th century after the irrigated areas were abandoned, to be restored much later. There are parallels for this: an Andalusi irrigation channel restored as late as 1623 was attested in Cieza (Siyāsa) (Navarro Palazón and Jiménez Castillo 2007, p. 99).

## 4. Arabic Sources

Although only two Arabic sources refer to the *almunia* of Castillejo de Monteagudo, they make it clear that during the Andalusi period, the complex was known as *Ḥiṣn al-Faraj*.[24]

The earliest mention is in Ibn Ṣāḥib al-Ṣalāt's (1987, p. 136) chronicle, which gives an account of the two destructions of the estate by the Almohads in 1165 and 1171. It seems obvious that for the Almohads to destroy the state a second time, Ibn Mardanīš must have had it restored or at least substantially repaired and/or altered. The archaeological excavations carried out in 2019 seem to confirm this, as the new palace shows a large extension that, it could be argued, represents works undertaken after the sack of 1165. In



his account of the later siege (1171), Ibn Ṣāḥib al-Ṣalāt explicitly claims that *Ḥiṣn al-Faraj* was Ibn Mardanīš's "pleasure estate":

"They came to Murcia, and besieged (*nāzalū*) and seized (*istaglabū*) Ḥiṣn al-Faraj, which was Ibn Mardanīš's pleasure estate (*mutanazzah*), and razed [its] walls (*marbāḍāt*) and orchards (*basātīn*), as well as the surrounding (dry) fields (*basāiṭ*)[25] and the villages (*qurà*) near the central (*mūsaṭ*) area (*balad*)."

The text reveals that the estate, in addition to walls, also had market gardens and farmsteads nearby. The mention of the villages as the dwelling place of the peasants who worked the land highlights the economic importance of the estate.

The second mention is in Ḥāzim al-Qarṭājannī's poem (al-Qarṭājannī 1925)[26] *al-Qaṣīda al-alfiya al-maqṣūra*, dated to the mid-13th century, shortly before the Castilian conquest:[27]

"¡And how many moments of joy (*furaj*) we had in the exalted (*al-sāmī*) Ḥiṣn al-Faraj, which drove away my sorrows until I was rid of them!"

The fact that the term *ḥiṣn* was generally used in al-Andalus to refer to hilltop fortifications explains why most researchers (Arabists and archaeologists) believe that this toponym alluded to the main fortress in the region, the Monteagudo Castle (Figure 4).[28] In all fairness, of all the Andalusi buildings in this area, none fit the word *ḥiṣn* like this one. Therefore, why did al-Qarṭājannī and, above all, Ibn Ṣāḥib al-Ṣalāt refer to Ibn Mardanīš's pleasure estate as *Ḥiṣn al-Faraj*? The Monteagudo Castle was the most prominent topographical reference in the region; it predated the royal estate and, when the latter was founded, the castle was rebuilt to be incorporated into, and preside over, the estate. It thus seems logical to assume that the whole (the estate) was named after one of its parts (the castle or *ḥiṣn*); synecdoche is a very common lexical phenomenon in toponymy. In fact, we know from Ibn al-Abbār (Ibn al-Abbār 1963–1964, II, p. 124) that in the 11th century, the Monteagudo Castle was called *Ḥiṣn Munt Aqūṭ*, a Mozarabic term transcribing the Romance toponym into Arabic (Munt = Mount, Aqūṭ = Agudo), which obviously links with the rocky spur on which it stands. The change from *Ḥiṣn Munt Aqūṭ* to *Ḥiṣn al-Faraj*, meaning "castle-belvedere" or even "castle of solace",[29] seems to have come about after the construction of the palatial estate, which is a better fit for this name than the castle.

Late medieval texts prove that the toponym did not disappear after the estate was broken up, as al-Faraj is often found as Larache in the north to refer to different geographical features: the plain, the headland, the *wadi*, the fortress, the watchtower, and the mountain range. This widespread use is an indication of the large size of the estate. Today, the toponym has disappeared and only survives in its Castilian form in the fortified residence of Larache (Figure 5); this is perhaps not a coincidence. As noted, this building is probably the only palace in the estate that remained in use at the time of the Castilian conquest in the 13th century,[30] which is why it may have been reused by King Alfonso X and granted to the bishops of Cartagena in the 14th century. In 1465, the palace was referred to as "castellar de Alabrache" (Molina Molina 1989, p. 291). This was the last mention of the palace in the record until the 18th century, when Joaquín Saurín turned it into his residence and museum (Lozano 1794, p. 165).

In addition to mentioning the estate by name, Ḥāzim al-Qarṭājannī describes several of its buildings. In the following paragraphs, all those that have been identified archaeologically will be described.

The first is *Muntaqūd* (verse 293), described as a place where the author and his friends took "pleasant walks […] far from vice and obscenity", which must refer to the area beneath the promontory, where the homonymous village is located today (Figure 3). It is not surprising that Ḥāzim al-Qarṭājannī mentions *al-Marj* (the marshes) in association with this toponym; as noted, an extensive area of marshland, referred to in the late medieval record as *almarjal of Monteagudo*, was part of the estate.

The second is the hill of *Kudyat al-Rašīd* (verse 298), cited as a hunting area, which proves that the estate included a mountainous sector in which animals roamed free. It is possible that a zoo also existed on the eastern slope of the hill on which the fortified

palace stands (Navarro Palazón and Jiménez Castillo 1995a, p. 97). This is based on the remains of a large enclosure, approximately 298 × 144 m in size. Keeping exotic animals is a well-documented practice in royal estates in Madīnat al-Zahrā', Marrakesh, and Tunis; animals were either confined in buildings or open-air enclosures or simply let loose in the gardens, provided that their presence was compatible with other uses.[31]

The third is *al-Ṣuhayrīj* (verse 482), a diminutive form that can be translated into "small pond". The smallest pond in the estate (Figure 3) is the one that presides over the pavilion in the palatine area of the plain, so it may be assumed that this is the one mentioned in the poem.

The fourth is *al-Burūj* (verse 480) which can be translated into "the towers". These could be the towers in the castle walls or in the Castillejo, although it seems more likely that the poet is referring to the towers that were scattered around the plain, from which the plain and the orchards, gardens, and even the marshland could be admired from above, like the belvedere tower excavated in 2018, of which more later. Written sources from the 13th century prove that this type of tower was common in these estates, and sometimes properties with a residential building are automatically described as towers (Jiménez Castillo 2018a, pp. 775–79; 2018b, pp. 59–62).

The fifth is *Qaṣr Ibn Sa'd* (verses 294, 297 and 479), which should be translated into "Palace of Ibn Sa'd". This is an obvious direct reference to Ibn Mardanīš, as further shown by the fact that the name features three times. In this instance, the toponym refers exclusively to the fortified residence (Castillejo) on the promontory (Figure 4), and not to the estate as a whole as previously thought. Its location in a prominent position seems to be proven by verse 479, which says that "The clouds delight (*as'adat*) the palace (*qaṣr*) of Ibn Sa'd when they rise from the slope (*munḥadar*) to the top (*mustamà*)". According to the poet, this was a splendid and evocative ruin whose "beauty had been erased by the misfortunes of a time long gone [...] in which time left a warning for those who remain". It is clear from the text that by around 1230, the palace was in ruins after the estate was destroyed by the Almohads. The study of the materials recovered during Sobejano's excavations in the Castillejo suggests that it was indeed destroyed and never restored.[32] In the 2018–2019 excavations undertaken in the plain, a similar sequence could be attested; the main palace was never rebuilt, but some minor 12th-century residences were restored and were in use at the time of the Christian conquest in the 13th century.

Al-Šarīf Al-Garnāṭī (1997, vol. II, pp. 657–58), supreme *cadi* of Granada, edited, annotated, and commented on the *Qaṣīda maqṣūra* in the 14th century. In addition to providing valuable information about Ibn Mardanīš, he indicates that *Qaṣr Ibn Sa'd* is named after him, which proves that it was Ibn Mardanīš who ordered its construction:

> "*Qaṣr Ibn Sa'd* is in Murcia and takes its name from Amir Abū 'Abd Allāh Muḥammad b. Sa'd b. Mardanīš al-Juḏāmī, [...] he seized the Cora of Tudmīr, and entered Murcia in the middle of *jumādà I* of 542. His vengeance (*sayara fī l-ṯu'ar*) was extraordinary, [and he was noted for his] great strength, bravery and solid [physical] constitution. Celebrated for his sagacity and chivalry, [Ibn Mardanīš], shared the table with great champions and raging warriors. Perhaps he shuddered (*hazza-hu*) to seek tranquillity and repose (*irtiyāḥ*) in its comforting residences (*fī majālis rāḥati-hi*); he preferred his diners [above all else], so he distanced himself from his bed and his children; and it was there that he used to lie with more than two hundred female slaves under one and only one blanket (*liḥāf wāḥid*)."

## 5. The *real* de Monteagudo in the Late Middle Ages

As in the Islamic period, the estate of Monteagudo must have been the property of the *majzén* (state). It seems logical to assume that it passed directly into the hands of the Castilian crown after the conquest in 1243. This is further suggested by fifteen royal documents issued there in 1257.

We know that the king gave at least part to Queen Doña Violante in 1266:

"In the *reyal* of Monteagudo, in the vineyard and in the dry land, the Queen has DC *tahullas*, which are XC *alffabas*"

and in the final summary of the third and fourth distributions:

"In Monteagudo the Queen has DC *taffullas*, which are XC *alffabas*, less the barren land" ([Torres Fontes 1960a](), pp. 1, 156).

As we do not know the estate's size with certainty, and as the information provided by the *Repartimiento* is meagre and incomplete, it is legitimate to ask whether this was the only *donadío* (600 *tahúllas*) and, therefore, whether these were the estate's total dimensions. Based on the *Repartimiento*, José Antonio Manzano argues that this was most likely not the case, pointing out that the solution lies in the third and fourth distributions, after the reference to the Queen's grant. The record indicates that the *partidor mayor*, Gil García de Azagra, received a *donadío* (p. 1) of 470 *tahúllas*, a total of 125 *alfabas*, in Cudiaçibit.[33] However, the following was recorded in the summary of these partitions, immediately following the xc *alffabas* presented to the Queen:

"Don Gil has there cccclxx *taffullas*, which are cxv alffabas" ([Torres Fontes 1960a](), p. 156).

The text shows that the *partidor*'s land was, like the queen's, in the *real* de Monteagudo, from which it is deduced that one of the two entries is wrong. The *donadío* in question could not be both in Cudiçibit and in Monteagudo. Which entry is correct? The answer seems to lie a few lines above (p. 156). The summary of the land awarded in the village of Cudiaçibit indicates that the *donadíos* granted there by the king amounted to 395 *tahúllas*, which seems to rule out García de Azagra's 470 *tahúllas*. In fact, the *tahúllas* in the *donadíos* granted in Cudiaçibit, excluding those granted to the *partidor mayor*, amount to approximately the 395 *tahúllas* mentioned in the summary, from which it can be deduced that the estate of García de Azagra was not in Cudiaçibit but in Monteagudo.

Since the queen's and García de Azagra's grants were part of the same estate, it is worth asking whether there were any other plots that were not included in the book of the *Repartimiento*. For this, it is necessary to examine the grant policy adopted for the *Repartimiento*.

*Donadíos* were valuable properties granted directly to those close to the king (relatives, courtiers, nobles) or to the military orders. These were different from the common grants awarded by lot, known as *heredades*. *Donadíos* were a common formula in the partition of properties that, because they had belonged to the Muslim *majzén*, had passed directly to the Crown after the conquest. A good example of this is the estate of the Alcazar Menor in Murcia, which was divided into several *reales*, such as "Queen Doña Violante's", "Infante Don Fernando's", "the partitioner's", and "the *adelantado* Alfonso García de Villamayor's". This was in addition to other minor properties, such as "Simon's orchard" and "Johan de Romay and Bernal Arens's houses". Interestingly, in this instance, the king kept the main houses and the associated orchards for himself ([Navarro Palazón and Jiménez Castillo 2011a]()). All this apparently amounted to some five and a half hectares, the equivalent to approximately 50 *tahúllas*, considerably less than in the Monteagudo estate, although in the Alcázar Menor, land was much more valuable.

It is, therefore, reasonable to argue that, in addition to the two *donadíos* recorded in the *Repartimiento*, there was a third part of the estate, which coincided with the areas likely to have been restored by Amir Ibn Hūd al-Mutawakkil (1228–1238) to replace the Mardanisid palaces destroyed by the Almohads, like in the Alcázar Seguir, in Murcia, where a succession of two palatial areas has been attested. The more recent was appropriated by the Castilian Crown and later turned into the monastery of Santa Clara la Real. A solid indication that the king kept part of the estate is that the royal court was based in Monteagudo in 1271–1272, years after the estate was subdivided.[34]

Until now, it was believed that Alfonso X took up residence in the Monteagudo Castle, but this did disregard the building's rundown state at the time or the large size of the king's itinerant entourage. Excavations found no remains of residential or palatial buildings but

many water tanks and silos, typical of a state granary, reconstructed in the Mardanisid period. As it is certain that the whole court was not housed there, it seems more logical to argue that all, or most of it, settled in the estate, where 13th-century buildings with levels dated to the Christian period have been identified.

Be that as it may, many of the characteristics and the size of the estate are still unclear. However, the available evidence makes it clear that it was relatively large, as the sum of the *donadíos* granted to the queen and the *partidor mayor* results in 1070 *tahúllas*, over twice the average size of the agricultural areas associated with village estates in the *huerta* of Murcia in the mid-13th century (523 *tahúllas* divided into small holdings with an average size of approximately 4 *tahúllas*) (Manzano Martínez 1999).

The description of the Monteagudo estate as "royal" in the third and fourth distributions is a matter of historiographical debate because, if its location and the extent of its dryland agricultural plots are taken into account, the term "rahal" seems more appropriate; in fact, the main road running across the estate, connecting Monteagudo with Cabezo de Abajo (Cabezo de Torres), probably the Alabrache road of 15th-century documents, is called *carril de la Almazara* or *senda "del Raal"* (Figure 3).

Castilian and Aragonese medieval sources use Arabic terms, such as "real" or "raal, rahal", to designate a specific type of rustic property; however, these words are currently in disuse and have lost their former semantic nuance (Jiménez Castillo 2018a, 2018b). Given the historical implications of their meaning, the issue has been paid considerable attention (Guichard 1979, pp. 17–20; 1982, pp. 45–46; 1989; 2001, pp. 504–22; Barceló Torres 1982, pp. 45–47; Rubiera Mata 1984; Glick 2007, pp. 48–49).

In his thesis, Pierre Guichard argued that, unlike the *qarya* (village), whose lands and houses were owned by many, the *rahal* belonged to a single individual and was often an aristocratic demesne that included a leisure residence and a farm. Because of the semantic similarity of "real" and "rahal", and the confusing way in which they were used in Christian sources, Guichard made no distinction between them, as pointed out by María Jesús Rubiera. She explained that the first term derives from *riyāḍ* (garden), and the second from *rahal*, whose translation, according to her, could be "sheepfold". That is, the word's primary meaning refers to stockbreeding, from which the word evolved to allude to a "farmhouse". According to Rubiera, "reales" were much more valuable than "rahales", and the term "rahal" gives no indication of ownership status but only of its use for stockbreeding.

Pierre Guichard eventually admitted that the words had different origins but rejected the interpretation of "rahal" as a farm used exclusively for stockbreeding. He concludes that both terms referred to the same type of private and aristocratic farm, with both residential buildings and cultivation areas. Furthermore, the toponymic analysis of *reales* and *rahales* shows that, while the former usually bear the names of specific personalities, the latter are frequently designated with institutional names (*rahal al-wazīr, rahal al-qāḍī, rahal al-mušrif*), sometimes in the company of a personal name, for example, the *raal* of *alcaid* Alpich. Therefore, based on the two main distinguishing features of *rahals*, their frequent connection with the state apparatus and their marginal geographical position, Guichard suggests that they were public lands temporarily granted to rulers, high officials, and army commanders (Guichard 2001, p. 521).

Concerning "reales", Rubiera argues, following Elías Terés, that *real* comes from the Arabic word *riyāḍ*, which in the Hispanic Arabic dialect meant "garden, orchard"; they were also regarded as very valuable properties. In the *repartimientos*, they were often granted to high-status people or were kept as *realengos*; for this reason and for reasons of homophony, they were sometimes called "*real/riyāḍ*", *regale* (Rubiera Mata 1984, pp. 120–21).

According to Guichard (1990, II, pp. 374–79; 2001, pp. 504–11), *reales* were aristocratic properties, including suburban orchards and important villages. They comprised a leisure house probably surrounded by a fenced garden. He also argues that they must not have been particularly large, as they were often included in larger grants.

Denis Menjot (2002, pp. 74–77), who studied the characteristics of Murcian *reales* based on the *Repartimiento*, argues that they are difficult to characterise and that they were scattered among the villages of the *huerta*. They were few, small, and tended to have a very high tax value, another clear difference with *rahales*, although both types usually had a single owner.

Likewise, Martínez Carrillo (1997) concluded, first, that Murcian *reales* formed a belt of small farms used to grow selected crops; second, that they were peri-urban in nature and were located inside and outside the suburbs; third, that they were supplied by the main irrigation channel and sometimes also by the Caravija irrigation channel; fourth, that they were enclosed properties; and fifth, that they belonged to members of the urban patriciate. Based on available documentation, they were enclosed by earthen walls or hedges to keep crops, tools, and animals safe.

According to the information conveyed by the *Repartimiento*, the *donadío* "real de la Reyna" comprised different types of land: vineyards, dry lands (*albar*),[35] and orchards, as well as unproductive "terra yerma". It bordered a sector of the marshes of Monteagudo, which was approximately 340 *tahúllas* in size and most of which (300 *tahúllas*) was granted to Guillem de Narbona (Torres Fontes 1960a, pp. 238–39). Grants of marshland in the *Repartimiento* suggest that they had some economic value, although this was probably small, as they are not counted in *alffabas*, so it can be deduced that they had no fiscal value. In the Andalusi period, however, the areas of woodland and marshland were part of the royal estate and may have been used for hunting. As such, the estate shared some characteristics with "rahales"—it was located in the periphery of the *huerta* and included large expanses of dry land—and also with "reales"—it had large orchards and a rich palatial area. In Manzano's opinion, the designation "real" can also be the result of semantic contamination by homophony (in Spanish, "real" also means "royal"). However, there are significant indications that the "real de la Reyna" was regarded more as a "rahal", as the road linking the modern towns of Monteagudo and Cabezo de Torres across the estate is still known as "Camino del Raal" (Figure 3).

The repopulation process changed the productive structure of the *huerta* of Murcia after the conquest. The poorest areas were depopulated, and some irrigated areas were abandoned or became lakes. This was the case in Monteagudo, where Ferdinand IV awarded 1000 *tahúllas* of marshland to Juan de la Peraleja in 1308 (Torres Fontes 1980, p. 88). By ordering the redistribution of land abandoned by their original owners, the monarch tried to mitigate flaws in the initial repopulation process, which resulted in many uncultivated, unpopulated, and flooded plots:

> "You should know that Pero Martinez Calviello and Berenguel de Pujalte, your lords, told me that your territory, in the dry lands as in the marshes of Monteagudo, and in other places, lie abandoned and barren, because its owners have left of their own free will. Some have been left without masters or cultivators because their owners have relocated elsewhere and do not live there any longer. If such properties were to be granted and divided by you, that would be my in service and would see the place repopulated".

## 6. The Palatine Area in the Plain

The "palatine area in the plain" comprises the group of architectural features found across the orchard area to the west of the mound on which the fortified palace known as Castillejo de Monteagudo was built. All of them are organised around a hypothetical transept garden, of which little is known (Figures 13 and 14).

The remains of the "aulic complex" are in the eastern sector of the transept garden and were discovered during the 2018 and 2019 excavations, which targeted the strip of orchard land planted with citrus trees below the Raal or Almazara road that connects Monteagudo with Cabezo de Torres. Traditionally, this road has separated the rain-fed and irrigated fields. Aerial photographs from 1929 show that this sector adapts to the winding road, the layout of which is different from that of the lower road that lies further west; its peculiar

course responds to the fact that it once ran on the edge of a wadi, which, in the mid-12th century or earlier, was filled in to prepare the ground for the great cross-shaped garden. This strip of land is 2 or 3 m higher than the old bed of the wadi to protect it from flooding.

This area was chosen to begin the archaeological exploration of the estate in 2018 because it is the sector with the largest density of known medieval architectural features, including some that were visible above ground level. Such a large concentration of remains in a strip of land some 170 m long could only be understood through excavation.

The two private plots affected by the excavation, although close to each other, are separated because the owner of the orchard in between did not grant the necessary permission (Figure 15). In 2018, the excavation was limited to the southernmost plot and revealed the foundations of a tower and two residential buildings organised around courtyards (Figure 16). In 2019, work focused on the northern property and exposed the remains of the pavilion-palace, portico, and pond. The medieval aqueduct, built over 12th-century constructions and still in use in 2019 as part of the traditional network of irrigation channels, lay between the two excavation areas. It is unclear how the structures in the two sectors relate to each other and to the northern pavilion (No. 14); however, we think that this aulic complex adopted the same layout as the palaces in al-Rummaniyya.

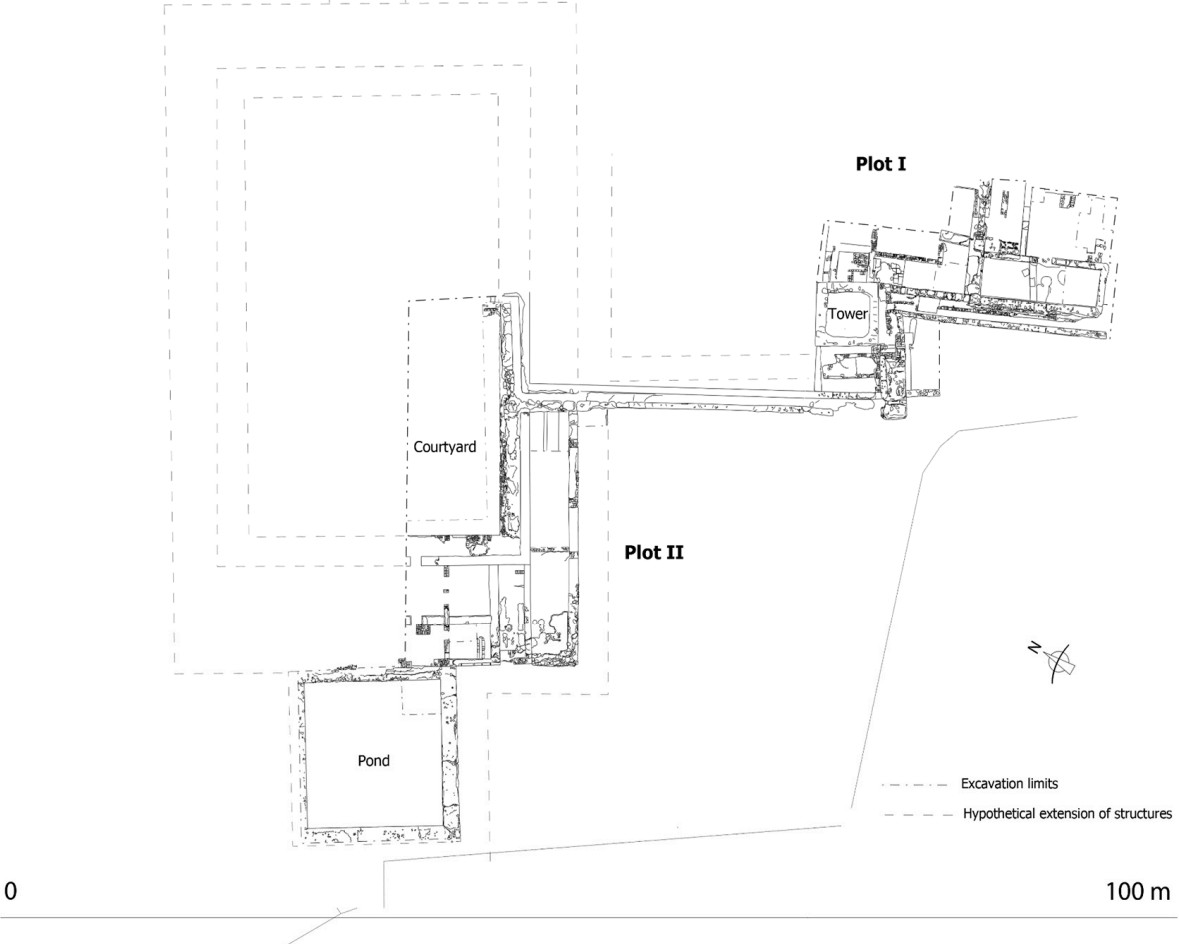

**Figure 15.** General plan of the palatial area: domestic area and tower in Plot I and the palace of the pool in Plot II, linked by the aqueduct.

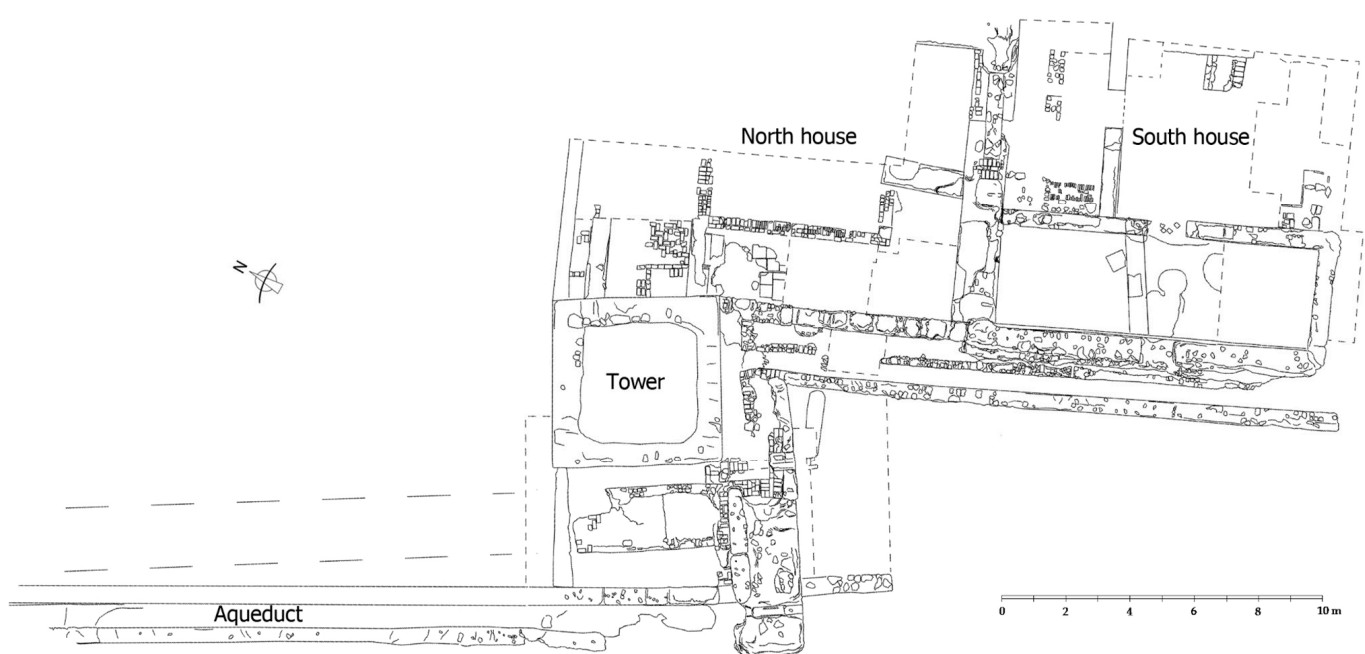

**Figure 16.** General plan of the domestic area, tower, and aqueduct (Plot I).

Although the area excavated to date only accounts for a small part of the whole complex, the results have completely changed existing notions about the estate. It was previously believed that the Castillejo was the only major residential building in the *almunia*, and that the land below was exclusively taken up by orchards and gardens and their corresponding hydraulic infrastructures, dotted with some minor constructions (e.g., the pavilions). However, based on the results of the excavation, the re-examination of the written record, and the analysis of field systems, it has become clear that most of the 12th-century palatine sector was in the plain, and that the pavilion-palace, excavated in 2019 at the centre of the aulic complex, presided over a large cross-shaped garden, with a 227-metre-long east–west axis (Figure 13).

As noted, the aulic complex must have adopted the same tripartite arrangement as al-Rummaniyya, as suggested by the three most outstanding buildings. The main one is the central pavilion-palace, which is flanked to the south by the tower exposed in 2018 and to the north, in the unexcavated area, by pavilion 14 (Figure 13). The east–west axes of the transept garden project from these constructions, so the architectural features were spatially aligned with the garden. In the hypothetical plan, the pavilion-palace and the pool project towards the centre of the garden, emphasising the transept's main axis, while the other buildings stand further back, stressing the secondary axes/transepts. This spatial subordination probably reflects functional differences, too; while the pavilion-palace was the most public and representational space, the others were likely restricted to more private or purely domestic activities.

Based on excavation results, it can be argued that the fortified palace of El Castillejo, in addition to playing a residential and protocol role, like the buildings in the plain, must have had very specific functions, such as serving as a lookout point and, at the same time, displaying power in the landscape, owing to its prominent hilltop location in the Murcian *huerta*.

## 6.1. The Tower

Given its location on the east–west axis of one of the garden's minor transepts, it is thought that the tower was a very important feature of the palatine area (Figures 16–18).

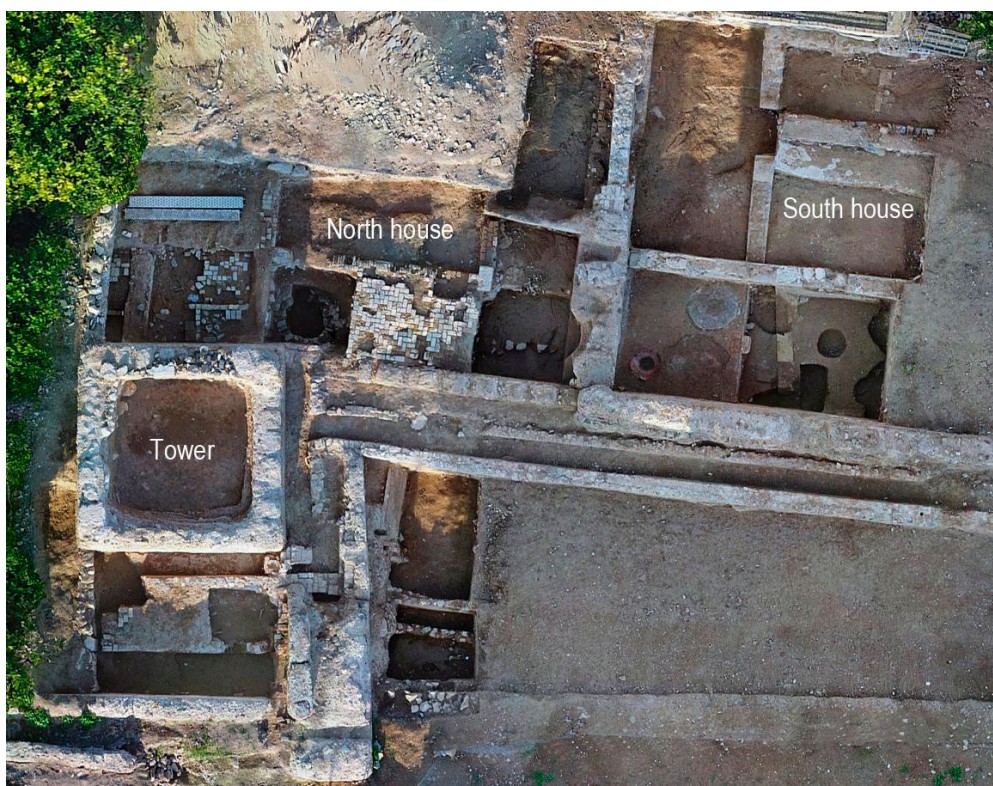

**Figure 17.** Plot I. Top view of the remains: on the left (north), the tower, the walkway, and the aqueduct; at the top (east), the remains of the two houses, separated from each other by a particularly wide wall.

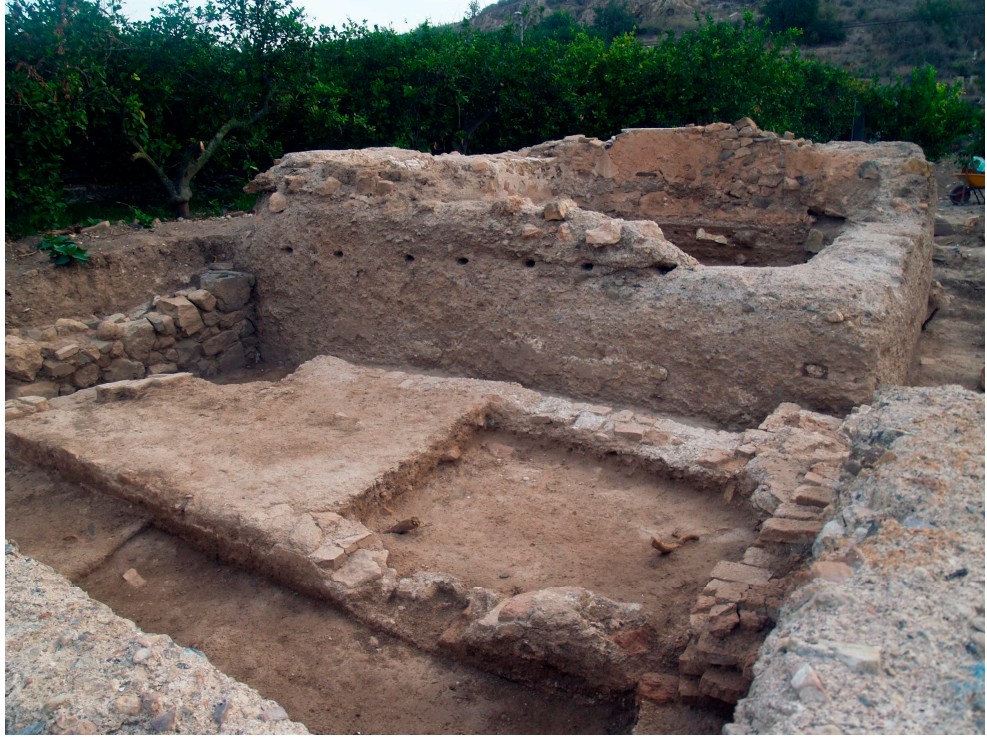

**Figure 18.** Plot I. Northwestern sector. Andalusian *tapiería* tower. Its solid interior was hollowed out in order to build an interior room in the modern period. In the back wall, there is a walled-up doorway belonging to the rural house that reused the tower.

It is square in plan (5.20 m × 5.20 m) and solid up to the preserved height (1.60 m). It was built with 75 cm-high rammed earth walls, the exterior face of which is lined with hard-packed lime and masonry, and the inside was filled with rammed earth.

The original height of the tower is uncertain, as it was demolished in the 1960s along with the house to which it was attached. The resulting debris, comprising large blocks of concrete walls, was documented in 2018 under the house's rubble. Its solid base was emptied of its earth filling and turned into a room, which proves that it had been reused. In addition, a 60 cm-wide doorway, later walled up with masonry, had been opened in the wall, and the internal faces of the walls had been roughly evened out with plaster (Figure 18).

Nothing is known about the superstructure, although it is assumed that the upper floor contained some kind of belvedere from which the estate could be viewed. This was probably accessed from the Andalusi building attached to its eastern face.

About 65 cm from the western face, there is a feature that appears to have belonged to a walkway, 1.95 m wide and 30 cm high (Figure 18). The walkway's walls are made of bricks taken with lime mortar and contain a filling of earth and small stones sealed by a brick floor. It is possible that this represents a later upgrade of an older pavement made with lime mortar, judging by the surface of this material found under the bricks, which is too smooth to be merely a bed for the bricks.

Coeval with the platform is a 65 cm-wide south-sloping channel between the western face of the tower and the walkway. It presents a 10 cm-thick lime mortar floor. Some traces of this feature survive along the southern face of the tower. The channel probably turned 90° to the south, abutting the wall that outlines the two residential buildings to the west (see below).

It is likely that both the channel and the walkway were designed to enhance the prominence of the tower, which presided over one of the minor transepts in the garden.

*6.2. The Domestic Area*

In addition to the tower, the excavations revealed two features arranged perpendicularly to each other, which, based on thickness and size, are regarded as the main ones in the area (Figures 16 and 17). One is a N–S wall on the step that separates the lower and higher terraces in the orchard. Two dwellings arranged around a central courtyard were found on the latter. Attached to the western side of the wall, there is a lime-lined brick channel that lies at a lower level than the aqueduct, which must have followed the same course, although it has not survived.

The second is a large E–W wall (1.30 m thick), which comes from the east to abutt the other wall. Its position and characteristics suggest that it separated spaces and buildings with different functions.

The excavation of the sector to the east of the first wall, on both sides of the second wall, revealed two occupation phases separated by a thick layer of rubble. It is very likely that the buildings attested on both sides of the wall did not initially have the same function, as suggested by the orientation of the only wall found in the southern sector, which is different from all the others. However, their chronology seems fairly homogeneous, and both buildings can be dated to the Mardanisid period, while their destruction is related to one of the two Almohad incursions (1165 or 1171). In the more recent phase, the buildings are less chronologically homogeneous, as it is thought that they were built over a longer time span, which could range from the late 12th to the mid-13th century. The following sections describe the two construction phases in both sectors in detail.

6.2.1. Northern Sector

To the north of the east–west wall, there is a dwelling whose earliest phase likely corresponds to the domestic and service area of the 12th-century palace complex. The layout is reminiscent of the buildings that flank the Hall of ʿAbd al-Raḥmān III, in the High Garden of the citadel of Madīnat al-Zahrāʾand the palatial rooms in the *almunia* of

al-Rummaniyya. The first and second construction phases are separated by a 90 cm-thick layer full of fragmentary remains of building material: mortar walls, bricks, tiles, stones, and sandstone slabs.

Early phase. As the walls and floor of the most recent house were still in place, the excavated area of the earliest dwelling was reduced to two soundings which, despite their small size, provided very significant information (Figures 19–22). The dwelling was organised around a courtyard with doorways to the north and south. Nothing is known about the eastern wall, which was not affected by the excavation.

- Courtyard. One of the soundings found the northwest corner of the courtyard, where a fragment of the original pavement survived. It was made of white sandstone slabs laid out at two different levels (Figure 19). The higher ones, next to the building's western formwork wall, are rectangular and are arranged perpendicularly to the wall, forming a perimeter pavement with reinforced foundations around the edges. The central lower slabs are laid out in N–S strips, forming a small step with the perimeter walkway.

On the western wall, which abuts the southeast corner of the tower, a fragment was found of the painted baseboard that must have run around the whole perimeter of the courtyard. The decoration displays a perimeter of rectangular *almagra* bands over a white background (Figure 20). It is possible that some of the panels were filled with geometric decoration, alternating plain and decorated panels, like in Castillejo de Monteagudo.

- Northern bay. It abuts the eastern face of the tower, and its interior was not excavated. All of its perimeter walls, except that to the east, were found. The north and south walls are made of rammed earth. The southern wall (78 cm thick) projects from the southeast corner of the tower and was cut through by the well of the now-gone modern house, exposing its interior structure.

A doorway (1 m wide, excluding the mouldings) links the bay and the courtyard (Figure 21). The doorjambs were neither reinforced with brick nor lined in lime mortar; however, the threshold consisted of a rectangular stone slab that lay higher than the courtyard walkway, forming a step. The interior of the room was paved with a smooth lime mortar floor, flush with the threshold.

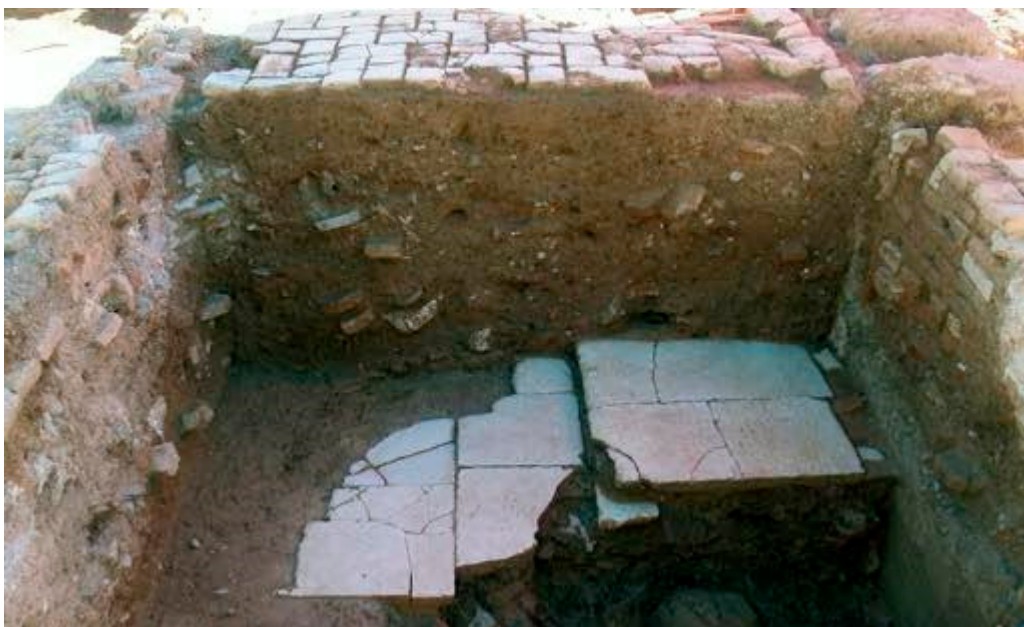

**Figure 19.** Plot I, northeastern sector from the north. The stratigraphic section shows the stone pavement of the early building, with the brick floor of the most recent above it. The demolition layer of the old building lies between both floors.

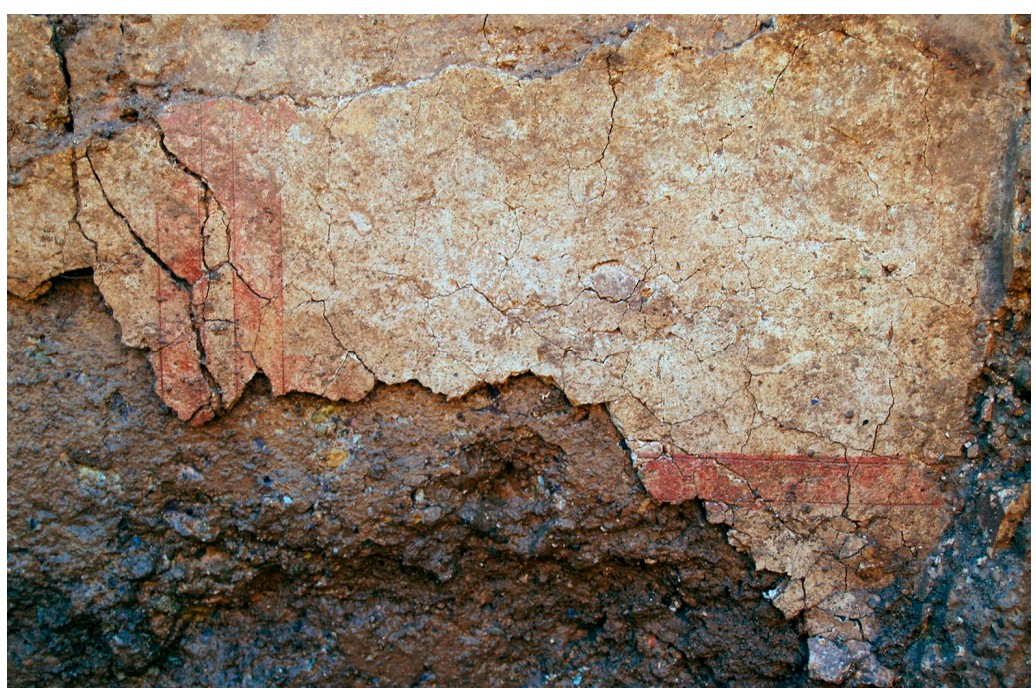

**Figure 20.** Old residential building. Courtyard. Detail of the painted baseboard on the northern end of the western wall, next to the tower. Mid-12th century.

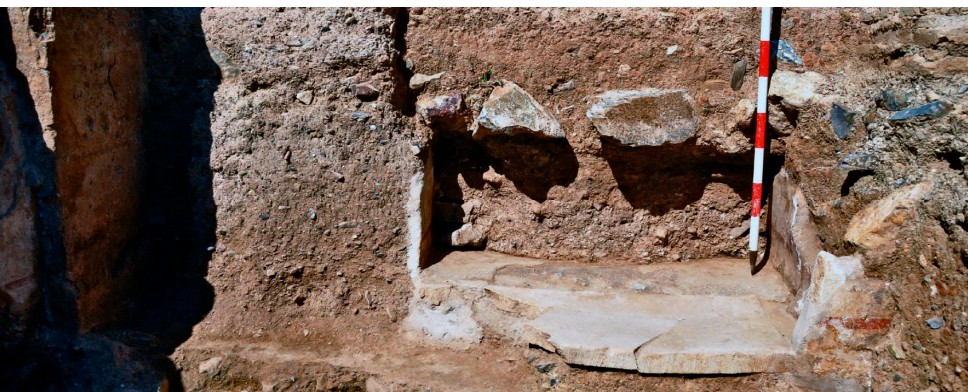

**Figure 21.** North house, old phase. Entrance door to the northern bay. On the right, it can be seen that the painted decoration that runs along the base of the courtyard continues along the jambs of the doorway. Mid-12th century.

The doorjambs were decorated with a geometric motif painted in *almagra*. Two horizontal strips converge and take right angles to run vertically in parallel, with a curved motif replacing the usual 90-degree angle. A more elaborated version of this motif is found in Castillejo de Monteagudo[36] and other Mardanisid monuments, such as the Dār aṣ-Ṣugrà[37] and the oratory of the Alcázar of Murcia (Figure 22).[38] A precedent can be found in one of the dwellings excavated under the Almoravid extension of the Qarawiyyin mosque in Fez, which is safely dated before 1134 (García Granados 2018).

-       Southern bay. Owing to the small size of the soundings, the wall that must have separated this corridor from the courtyard was not located, although there are some indications of its existence. The bay was paved with lime mortar, and some kind of raised masonry feature abutted the wall; this is the canonical position for alcoves or *alhanías*, but in this instance, the space is too shallow, so this must be ruled out.

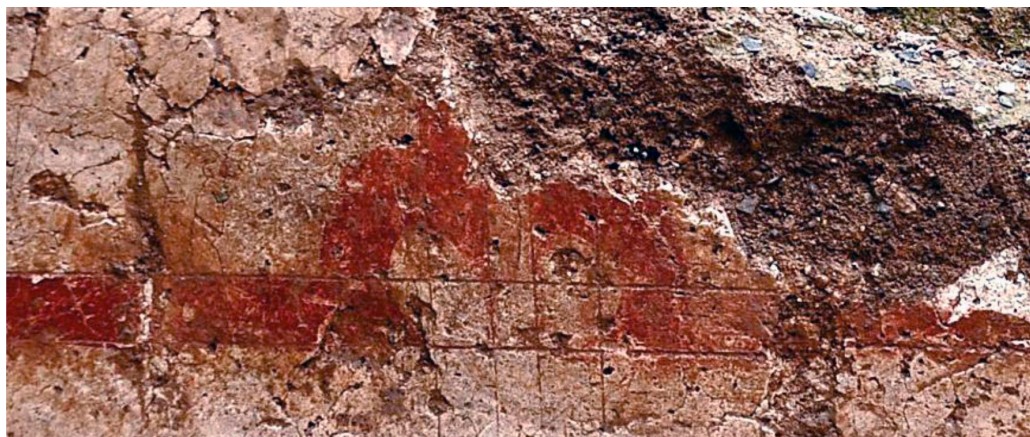

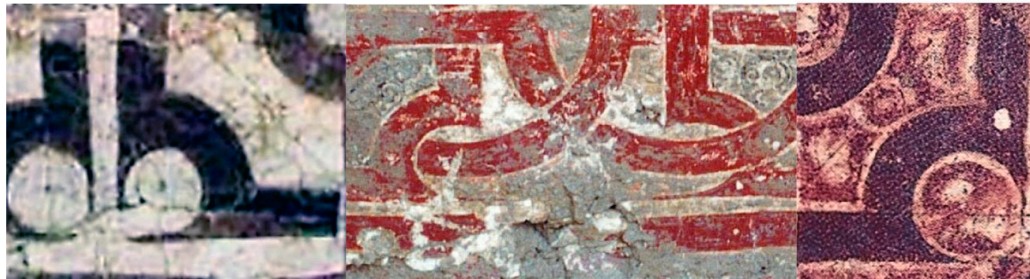

**Figure 22.** North house, early phase. Northern bay. Painted baseboards. The three lower images are parallels of the circle-motif: the left and centre images correspond, respectively, to the first and second decorative phases in the Dār aṣ-Ṣugrà; the right image corresponds to a dwelling under the Qarawiyyīn mosque in Fez.

Late phase. The new building was built upon a 120 cm-thick layer of rubble sitting on the collapse of the 12th-century house. Although different in some aspects, the new building was also organised around a central courtyard, largely because some of the original structures were reused as foundations for the new ones. The more recent house was only very partially investigated, as only the western bay, part of the southern bay, and a small section of the northern bay were fully excavated. Until the excavation can be brought farther to the east, the existence of an eastern bay will remain hypothetical.

- Western bay (Figure 23). It is rectangular in shape and was built over part of the earlier courtyard, reusing its west wall (which is very robust) as a foundation.

Only the northern and eastern walls have survived. The walls are made of rammed earth, with the southern one sitting on a brick foundation. Bricks were also used to reinforce the corners of the walls around the courtyard.

The brick pavement is partially preserved (the northern end is missing), and there are no breaks to indicate the presence of an alcove.[39] Little can be said about the opposite end, which was cut through by a later trench; however, it is likely that this room did not continue to the south, as a room in the southern bay extends into this area all the way to the western boundary of the plot. The thresholds of the double doorway are also made of brick, laid out in a fishbone pattern.

The location and morphology of this space suggest that it was a secondary hall, probably without alcoves, which opens onto the courtyard to the west through a double entrance divided by a central brick pillar. Double openings were a secular architectural solution that remained in use until the Mardanisid and Almohad periods, to be replaced later by single openings (usually forming a round arch). This change, which must have taken place around the 1220s, is well documented in the proto-Nasrid architecture of eastern al-Andalus (Navarro Palazón 1991a, pp. 24–25), e.g., the Hudi palace of Santa Clara, Murcia (Navarro Palazón 1995), Siyāsa (Navarro Palazón 1991b), and Onda (Navarro

Palazón and Jiménez Castillo 1995b). It should be noted that the doorways of the southern house described below already adopt the later solution, which suggests that it is more recent.

Although it is thought that this bay was not preceded by a portico, generally, porticos are located on a north–south axis in front of the main hall, which usually sits in the northern bay. Until this area is excavated, any reconstruction will remain purely hypothetical.

- Southern bay. The south wall of this bay is the large east–west wall that separates this house from the one to the south. The excavated area revealed two rooms separated by a wall, which is too thick to be a mere parting wall.

The westernmost room was enclosed on three sides (south, east, and west) by rammed-earth walls from the earlier construction phase. The state of preservation of the two faces of the north wall is very different; the western face has disappeared entirely, while the one facing the courtyard presents the same masonry technique as the wall of the western bay with which it forms an angle.

The easternmost room was also closed on at least three sides (south, east, and west) by formwork walls from the earliest phase. Nothing is known of the northern wall facing the courtyard, as its hypothetical location is outside the excavation area.

- Northern bay. Based on its location to the north of the building, this bay should be the main hall. The excavated area is at the base of the eastern face of the tower, built during the first phase. The north and south walls are made of limestone blocks mixed with abundant earth. It presents the same brick floor as the western room, and next to the base of the tower, there are two brick partitions that form a T-shaped feature, which is too shallow to belong to an alcove.

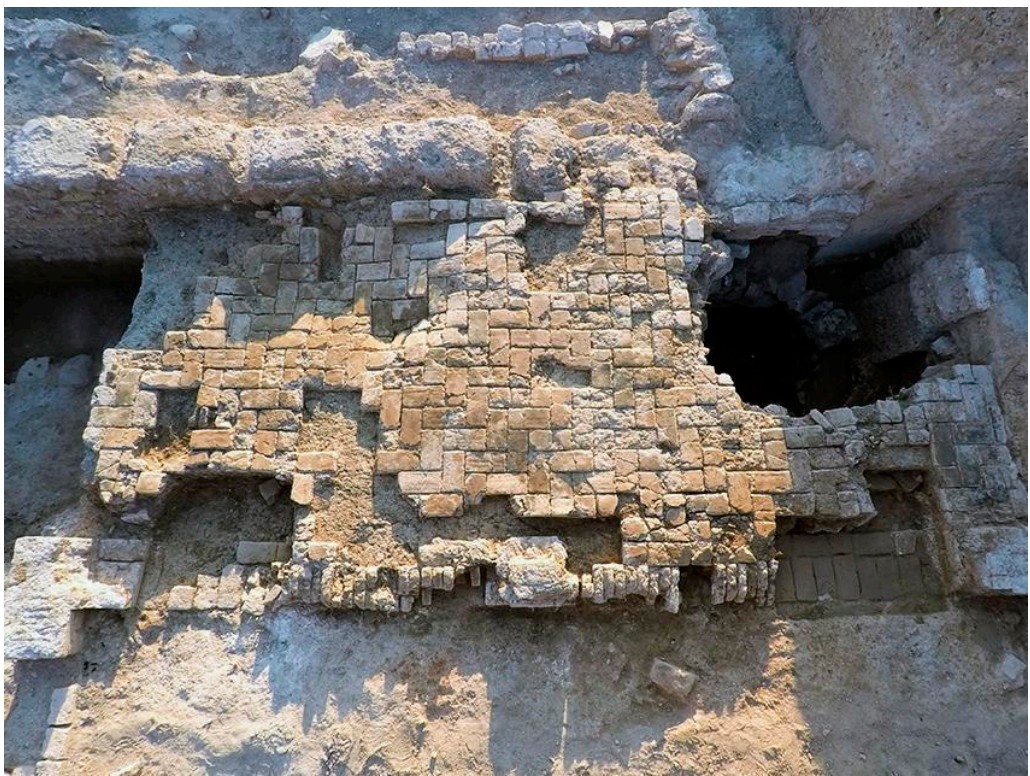

**Figure 23.** North house, late phase. Aerial view of the western hall.

6.2.2. Southern Sector

The next sector is located to the south of the large wall that cuts across this area. Although two clearly differentiated construction phases roughly equivalent to those in the northern sector have been identified, it should be pointed out that little information for the earlier phase exists; the only well-documented wall presents a different orientation to that

of the rest, making the domestic nature of the building uncertain. With regard to the most recent phase, the dwelling organised around the central courtyard is different from the one in the northern sector in terms of construction technique, suggesting it is more recent (Figure 24).

- Early phase. The only evidence available for this phase is a wall whose orientation is different from that of all the rest, which also abuts the wall that runs south to north across this area (Figure 24). The wall was built in formwork with an earth-rich fill and plaster on both sides; the southern face presents remains of painted decoration. A doorway connects two spaces whose lime mortar floors are at the same level as the early-phase pavements of the dwelling found in the excavation area to the north. The fact that the calcarenite slab that sits as a threshold is aligned with the northern face of the wall and that the lime pavement of the south room continues upon the doorway, strongly suggests that this pavement also covered the north room, although it is unclear if the latter was a courtyard.

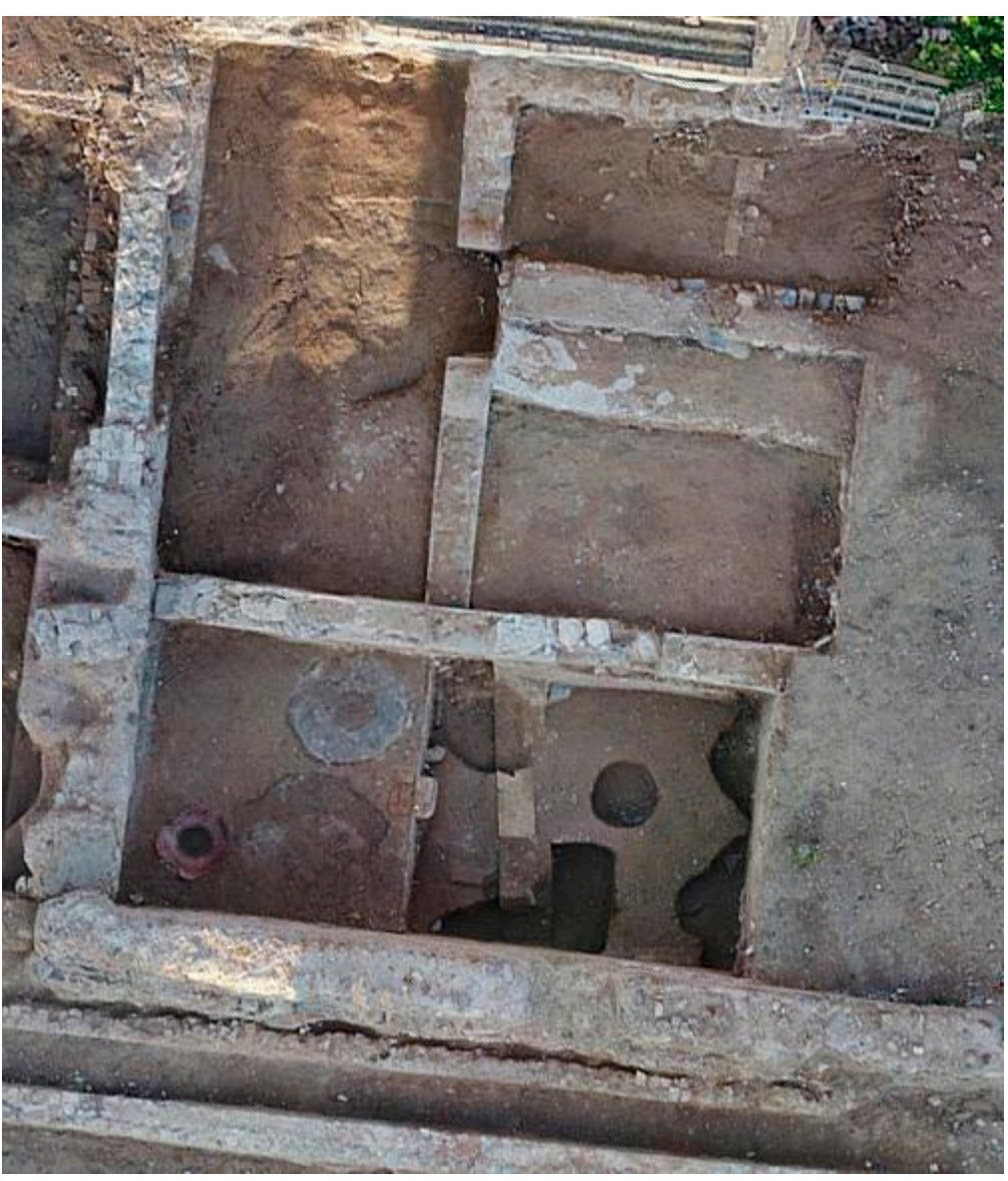

**Figure 24.** South house, recent phase. The survey shows the old phase of the 12th century. Zenithal view.

Interestingly, the doorways found in the south and north excavation areas are very similar. In both instances, the doorjambs are made of the same earthen formwork, and neither are reinforced with brick buttresses; similarly, their thresholds have the same calcarenite slabs. All of this and the fact that the pavements in both sectors are approximately at the same level suggest that the buildings are contemporary; based on the type of masonry used and the stratigraphy, they could be dated to the Mardanisid period.

- Late phase. The late phase of this excavation sector corresponds to a dwelling. The excavation revealed the central courtyard, a large room to the west, and a smaller room to the north (Figure 24). Nothing can be said about the other bays (east and south) except that the southern wall of the courtyard could be closing the building in this direction. The walls are made of rammed earth and the doorways have no central partition, which suggests that they postdate the twin doorway found in the late phase of the excavation sector to the north.

Although the house's floor plan is only partially known, it seems that the western bay was chosen to host the main hall, instead of the more usual northern location. This 'anomaly' may be due to the fact that the ground slopes from east to west. As the building's interior layout is uncertain, it is unknown whether it had any *alhanías*. The stratigraphic sequence suggests that it was occupied from the 13th to the 20th century. The medieval hall was later turned into a storeroom, as indicated by the presence of storage jars and pits cutting into the medieval floors; the most recent jars have been dated to the second half of the 19th century or the early 20th century. Although the archaeological sequence is correspondingly altered, the sequence of pavements and their marks on the wall plaster outline the different occupation phases.

The occupation of the courtyard also spans from the 13th to the second half of the 20th century, when the last house to be built there was demolished. The remains of a walkway were found near the surface in the southwest corner of the courtyard at the foot of the wall of the western bay.

At a lower level, in the eastern sector of the courtyard, remains of further architectural features were found. The bricks used in these features are different, in terms of size and fabric, from those found in the building in the northern sector; in fact, they also differ from the Andalusi bricks from the late 12th and first half of the 13th century found in the city of Murcia. In principle, these bricks date to the second half of the 13th century, when the estate was in Castilian hands.

*6.3. The Pavilion-Palace*

Although not fully excavated, a first approximation of the floor plan and architectural features of the pavilion-palace can be presented. Again, two construction phases could be clearly distinguished: the foundational phase, which comprises an undivided back courtyard, and a later phase, in which the courtyard was subdivided with parting walls, perpendicular to the original perimeter walls (Figures 25–28).

Courtyard

**Plot II**

Pond

- ·—·— Excavation limits
- ———— Hypothetical extension of structures

0

50 m

**Figure 25.** Plot II. Plan of the palatial area near the cistern, excavated in 2019.

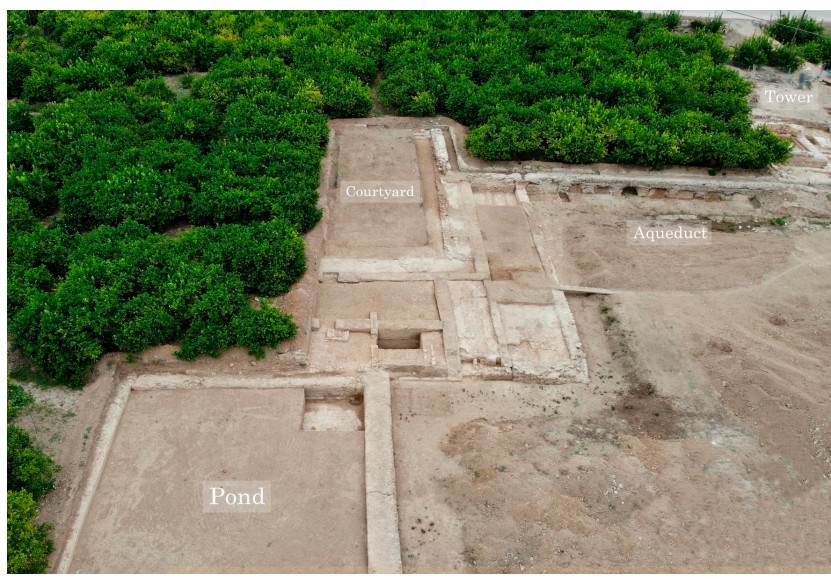

**Figure 26.** Plot II. Palace of the cistern, general view from the west. Drone flight on 21 October 2019.

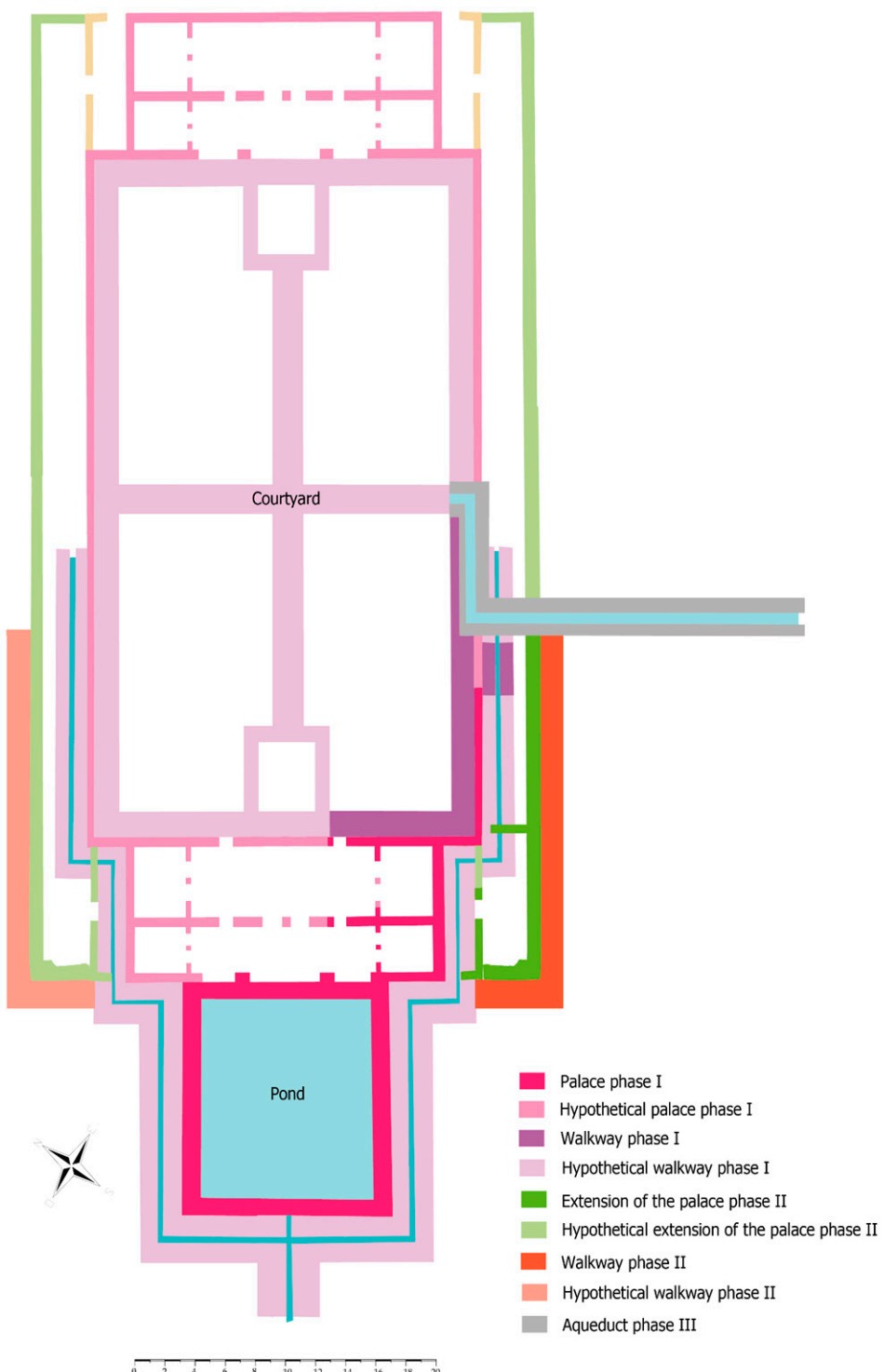

**Figure 27.** Plot II. Palace. Hypothetical reconstruction of the plan in all phases.

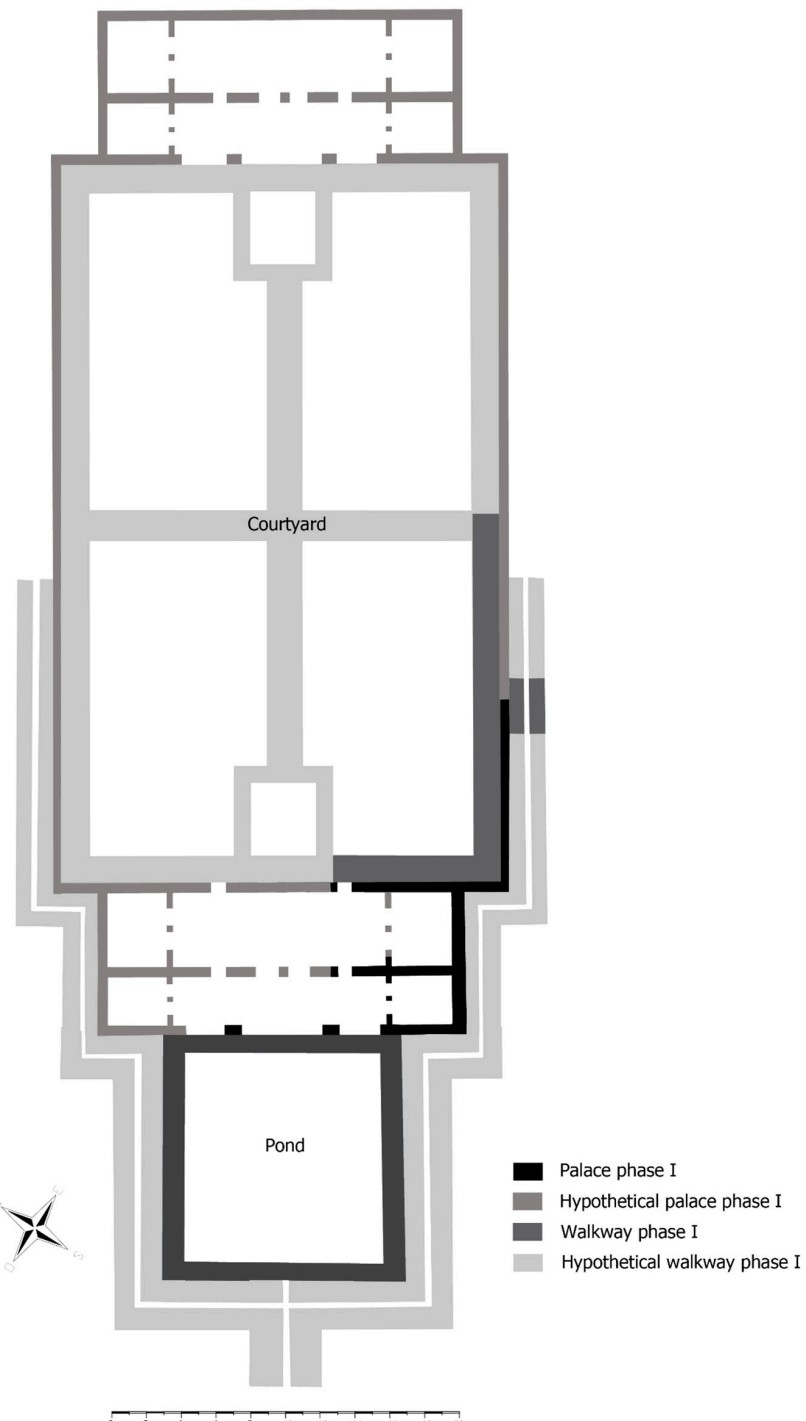

**Figure 28.** Castillejo de Monteagudo. Hypothetical plan of the palace in plot II. Foundational phase.

6.3.1. Foundational Phase

The study of a building of which only a small part is known is always challenging, but parallels—especially the *almunia* of al-Rummaniyya (10th century) and the Dār as-Suġrà (10th century)—as well the general architectural model can help. However, only future excavations can confirm or rule out the hypothetical reconstruction presented here.

- Cistern. It is a key element in the complex (Figure 26). Its floor plan is almost perfectly square, with interior dimensions of 11.95 m (E–W) by 11.30 m (N–S) (total size: 135 m$^2$); it is 0.90 m deep, resulting in a total capacity of approximately 120 m$^3$ or 120,000 litres. The walls, 1.30 m thick, are made of solid concrete lined with masonry, and the floor is

a thick layer of mortar. The eastern wall of the cistern presents a cut, which suggests that water entered the cistern from the south.

As well as acting as a water reservoir for irrigation, the aesthetic value of the cistern, which faced the portico and would reflect its architecture like a mirror, and the illumination of the interior through reflected sun rays, should not be underestimated.

- Portico. Only the southern third of the portico and the whole adjoining alcove could be excavated (Figure 29). The pavements were completely stolen. The stratigraphic analysis of the northern section of the portico suggests that the missing floor was approximately 25 cm thick.

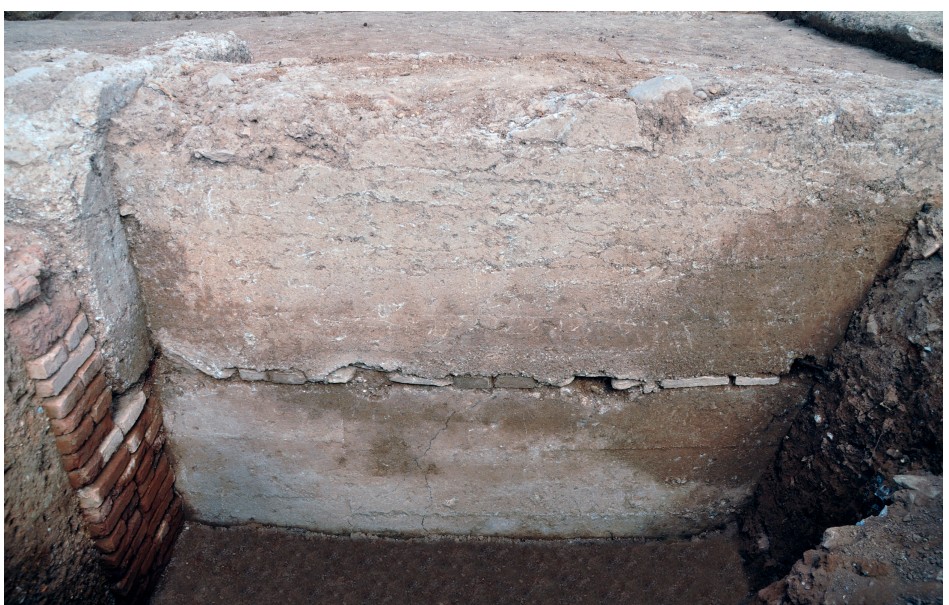

**Figure 29.** Plot II. Palace of the *pond*. Alcove in the portico. Stratigraphic survey. View from the west of the formwork foundation of the eastern wall. The separation of the walls with a row of bricks points to the construction technique used.

The only surviving remains of the arcade that adjoined the cistern are the foundations of two rectangular pillars on either side of the central opening and the southern jamb of the southern opening, all of which were made of brick (12 × 24 cm). Although the remains of the arcade are very fragmentary, it is believed that it had three arches.

A narrow doorway (0.88 m wide) was found in the southern third of the portico's eastern wall, leading to the main hall. The doorjambs were reinforced with a lime-rich mortar rather than brick buttresses (Figure 30). Presumably, there was an identical door at the opposite end, flanking the main door, which was situated in the building's central axis; these tripartite arrangements are known in Caliphate- and Taifa-period palaces.[40]

The portico's alcove is rectangular in plan (2.85 m wide and 3.55 m long). This anomaly is the result of the inclusion of two equally wide alcoves in bays of different widths; in this instance, the builders gave priority to the proportions of the hall over those of the portico.

The portico was connected to the alcove through a double doorway with a central partition. The bed of the threshold slabs has survived. In this type of architecture, the floor is generally lower in the portico than in the alcove. The latter presents rammed earth walls of varying thickness on three sides (S: 0.75 m, E: 0.80 m, and W: 0.62 m); the eastern wall is thicker because it is shared with the hall, which must have been higher than the portico.

A test pit (1.50 × 2.30 m) in the eastern half of the alcove, in contact with the wall, exposed the wall's foundations, which consisted of two superimposed walls separated by a brick wall (Figure 29). The pit also revealed that the building sits on a gentle east–west slope and that a terrace was built to keep all rooms at the same level.

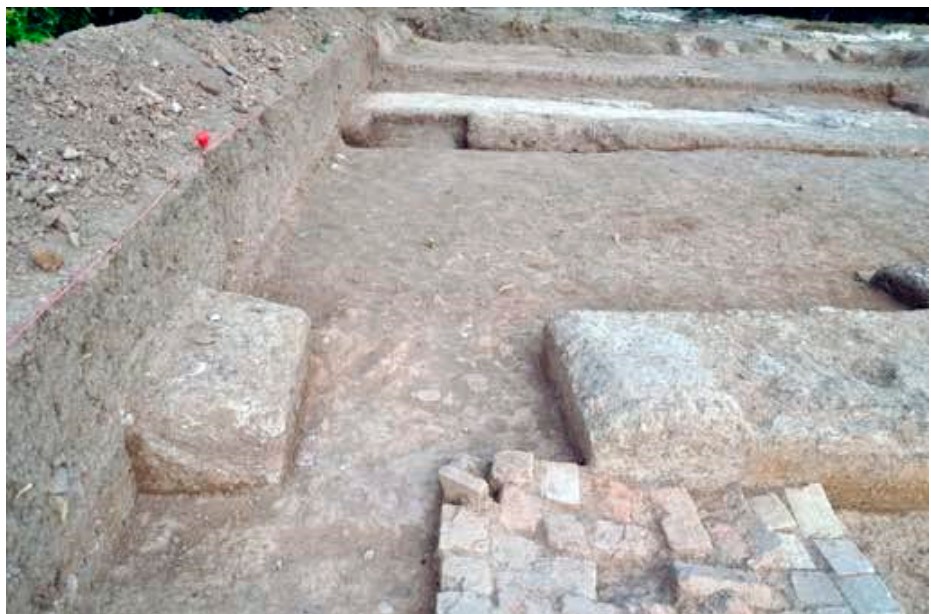

**Figure 30.** Plot II. Palace of the pond from the west. In the foreground, the opening that connects the portico with the hall; in the right-hand jamb, the formwork wall was reinforced with lime mortar. In the background, the door between the patio and the hall.

The foundations sit directly on the natural terrain; the constructors lay one course at a time, filling the interior of the walls with very clean earth, which was thoroughly compacted before the following course was laid. This technique is reflected in the sections in which thick homogeneous layers of earth alternate with much thinner layers. The latter, a by-product of form-working, corresponds with the top surface of each course, which also became the platform on which the following course was laid.

- Hall and alcove. Like in the portico, only the southern third of the hall and the whole of the adjoining alcove were excavated (Figure 29). Here, too, the floors, which were likely higher than those of the portico, are missing. Everything suggests that this space was thoroughly dismantled, including the floors.

The alcove is rectangular in plan (3.50 m deep × 4.10 m wide) and opens to the hall through a double doorway with a central partition. It has rammed-earth walls (0.77 m thick) to the south, east, and west, two of which it shares with the portico's alcove.

A doorway (0.90 m wide) opens in the front wall of the hall. It is off-centre with regard to the room's axis, but it is perfectly aligned with one of the doors into the portico; moreover, both doorways have exactly the same span (Figure 30). Assuming a typical symmetrical plan, it can be deduced that an identical opening existed on the opposite side, connecting the hall and the courtyard at the rear. As only the southern third of the hall has been excavated, we cannot exclude the possibility of a central opening to the rear courtyard. However, this is unlikely because this doorway would take up the focal point where the sovereign's seat or throne was placed during receptions (Navarro Palazón et al. 2018b, pp. 514–15), thus invalidating the room as a reception hall. However, even if the hall only had two small doors in the back wall and the central section was closed (Figure 28), the arrangement of this hall is still atypical because, as a rule, Islamic palatial reception halls are cul-de-sac spaces. Some exceptions, however, can be noted, such as the Dār al-Mulk and the House of Ja'far, both in Madīnat al-Zahrā'. It is thought that the two narrow doorways to the garden at the rear are explained by the fact that the pavilion projected into the garden, with the other three sides closed to the aulic complex. Therefore, this was the only way to open an interior communication route between the pavilion and the other aulic rooms, allowing the sovereign to exit the pavilion without going through the portico and around the exterior walkways to re-enter the palace.

- Courtyard. The courtyard behind this room, less than one-quarter of which has been excavated, is surrounded by a simple perimeter wall (Figures 25, 26, and 28). Until the excavation can proceed farther east, it will be impossible to say whether it opens to more rooms. However, in similar complexes, such as the palatial areas of al-Rummaniya and the Dār aṣ-Ṣuġrà (also the work of Ibn Mardanīš in Murcia), these courtyards often take a central position in palatial complexes, connecting directly with some of the most important rooms.

Like the rest of the palace's structures, the courtyard's southern wall was razed to the ground. It only survives as a section of formwork, 75 cm thick, so it can be deduced that it was fairly tall. If a symmetrical plan is assumed, it can be argued that a similar enclosure existed on the other side of the building, which does not exclude the possibility that the courtyard's eastern section was flanked by a number of rooms.

The excavation of the southern and western sides revealed sections of the walkways that ran around the perimeter of the courtyard, but these were heavily robbed out and damaged by ploughing.

A small section of the southern walkway reveals that these paths were paved with a thick layer of lime mortar inside a curb, most likely made with stone slabs. This sector of the walkway has survived better because it was sheltered by the construction of the aqueduct after the destruction of the palace.

Owing to the small size of the excavated area, it is not known whether the deep garden had a transept, a typical layout in 12th-century palaces. However, the 90-degree angles taken by the aqueduct across the courtyard suggest that this was the case. As noted, one of the aqueduct sections sat on the southern walkway, and the fact that the north–south arm of the deep garden's transept lies at its base suggests that it turned north further east.

- Outer walkway. The outside of the pavilion, which adjoined the cross-shaped garden, was surrounded by a perimeter walkway. The walkway was removed when a new bay was built to the south (Figure 31). This was confirmed by a sounding, which revealed that the walkway was approximately 2.50 m wide and that a channel, about 24 cm wide, ran across it. It presents a solid, well-finished lime mortar pavement. Its height in relation to the surrounding ground is uncertain.

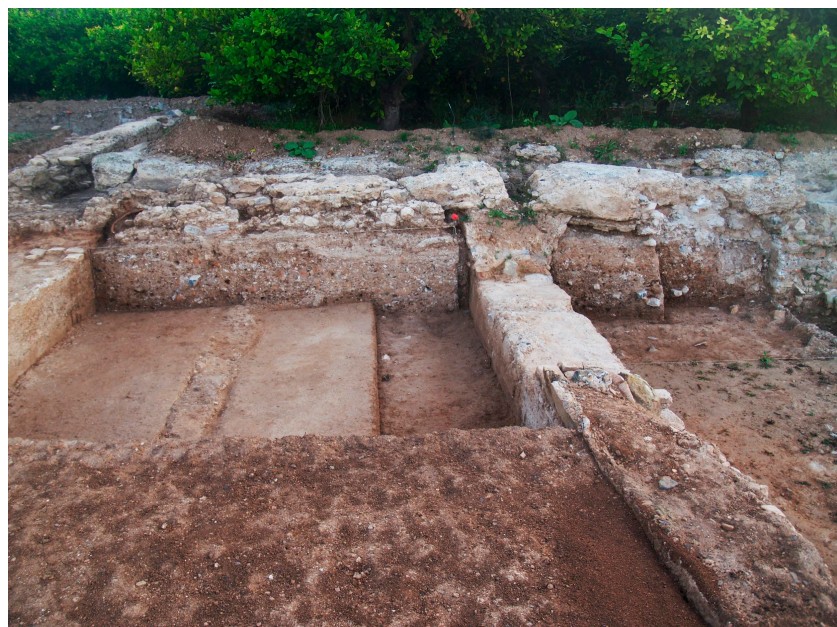

**Figure 31.** Plot II. Trench at the foot of the aqueduct. On the left, the southern wall of the palace in the second phase. Under the aqueduct, the remains of the walkway that surrounded the palace in its final phase are preserved in this section. The brick pillar on the right, the first in the aqueduct arcade, was built upon the walkway after the palace was in ruins.

### 6.3.2. The Refurbishment of the Palace

The renovation of the foundational complex involved the construction of two new bays, which abutted the wall that surrounded the courtyard and the short sides of the pavilion. This substantially altered the layout of the three main features: the pond, the portico-lounge, and the courtyard (a square and two rectangles of different widths, respectively, forming the projecting corners that characterise Mardanisid architecture) (Figure 27).

- Entrance hall. The extension of the palace to the south created room for the construction of a complex hallway in the southwest corner, which apparently connected the inner courtyard directly with the walkway built around the basin. The hallway must have been divided into three distinct sections: a relatively narrow and long corridor (7.80 m × 2.00 m) immediately inside the front entrance; a rectangular room, connected with the corridor through a doorway, which formed the corner of the building; and a room to the east, open to both the corner room and the courtyard. As the layout of this type of architecture is often determined by lighting requirements, the second room may have had an open roof, although this is purely hypothetical. Triple-angled entrance halls are not exceptional and can be found, for example, in the Palacio de los Leones in the Alhambra.

Remains of red painted baseboards were found in the bottom section of the southern wall of the corridor, near the entrance, in relation to a sound lime mortar pavement. The state of preservation of these elements suggests that the pavement in the corridor was relatively low, ascending by means of ramps and/or steps into the other rooms to reach, eventually, the level of the courtyard walkways. An east–west drain, which may have come from the southwest corner of the courtyard garden, ran under the pavement of the corridor; it is possible that the drain also flowed into a hypothetical latrine located at the eastern end of the corridor.

- Southern bay. This bay was not part of the original layout, and it was in fact built over the exterior walkways (Figure 31). It was 3.25 m wide, but its length is unknown because it continues into one of the plots that could not be excavated. It abuts the southern face of the palace, reusing its south wall. The new façade was made of rammed-earth walls on a plinth reinforced with added lime and stone. The only difference between the walls of the first and second phases is that the upper sections of the latter are reinforced by brick pillars.

As the building was heavily damaged, even below ground level, no remains of the doorways that connected the rooms in the southern bay with the courtyard have survived. Similarly, no sign of the walls that must have divided this bay into different rooms could be found, apart from the one that separated the first and second sections of the hallway.

- Second walkway. As noted, a walkway skirted the outside of the original palace, but this was built over with the addition of the southern bay. This led to the construction of a new walkway, the elevation of which has not survived, although a section is preserved under the head of the later aqueduct (Figure 31). The base of the walkway was supported by a single-face formwork retaining wall to the south, the core of which was a compact fill of earth mixed with architectural fragments, including bricks, roof tiles, stone slabs, mortar walls, etc.

### 6.4. The Aqueduct and the Abandonment of the Palatine Area

The medieval aqueduct remained in use until 2019 as part of the traditional irrigation network. It is the only architectural feature that could be examined between the area excavated in 2018 and the pavilion. The evidence suggests that the aqueduct was built over the 12th-century aulic complex, which was already in ruins. This indicates that the status of this area changed radically from residential to purely agricultural, which does not mean that the estate was broken up. The structural quality of the aqueduct suggests that it was not built by peasants. It consists of three sections, which will be described below, beginning from the north.

The first section is a solid structure of concrete and small stones. It sits directly on the pavement of the southern walkway in the pavilion's rear courtyard, running from east to west, which unequivocally proves that it was built after the palace was in ruins (Figure 31). The two walls of the channel were embedded in the ground, so only their interior faces were visible. After running about 10 m to the west, it turns to the south at a perfect right angle.

The second and longest section, which runs north–south, is approximately 32 m long (Figures 31–33). It flies over the drop that separates the palace from the crossed-shaped garden, so it is partially raised upon brick arches with different (and sometimes misshapen) profiles, including semicircles and ellipses. The arches have a span of 1.30 m and are supported by 80 cm-wide pillars. The builders took advantage of the uneven ground to make the aqueduct rest upon the 12th-century structures.

The two ends of this section do not rest on the arcade but on a set of pre-existing structures, which have been identified, for the most part, as the palace's walkways. The initial section of the aqueduct crosses the southern bay of the courtyard, smothering its walls. The added pressure made the courtyard's southern wall burst (Figure 31). Outside the palace, the bed of the aqueduct is heavily cracked because the walkway on which it sits was eventually removed as a result of agricultural activity.

At this point, a brick pillar, which was attached to the south side of the walkway, marks the beginning of the nine-arch arcade. Structural analysis suggests that the pillars and arches were built first, followed by the application of a thick layer of concrete and the construction of the channel and its side walls. In order to achieve the necessary slope towards the south, the threads around the arches' keystones become gradually thinner (Figure 32).

The same construction technique is found again beneath the ninth arch. In this instance, the arcade-end rests on a solid mortar block that sits upon the northern face of the main walkway of one of the minor transepts, which heads west from the tower. The eventual ruin of this feature as a result of agricultural work undermined the aqueduct's stability. Based on the imprint that it left on the hydraulic structure, the walkway was 4 m wide.

After this walkway, the aqueduct continues N–S for just two metres, where it takes a 90-degree turn. Like the first, the third section runs W–E (Figure 17) and ends in the southwest corner of the tower.

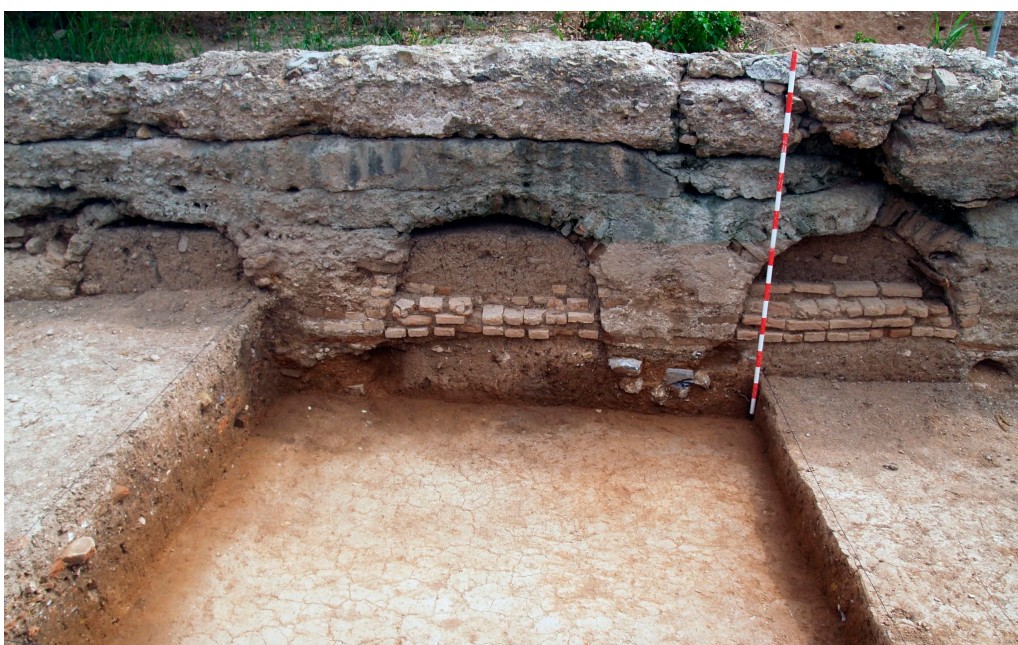

**Figure 32.** Plot II. Trench next to the aqueduct. Brickwork used to reinforce the arcade.

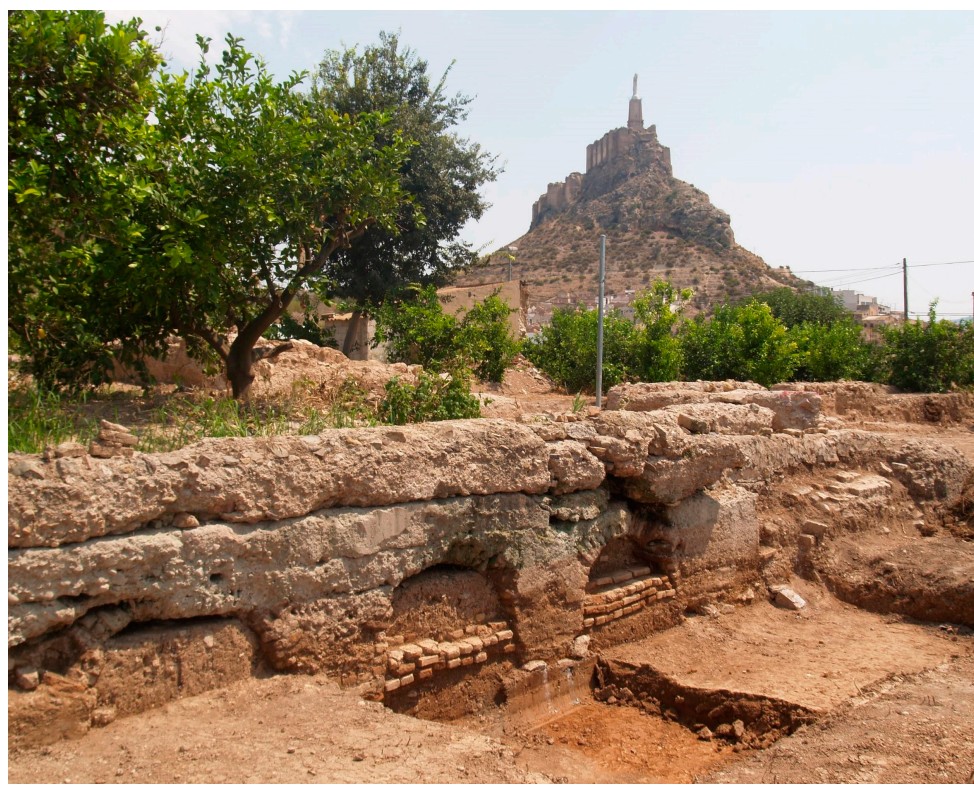

**Figure 33.** Plot I. South end of the aqueduct, with the devastated remains of the E–W walkway that began at the tower.

Based on the available chronological indications, most of which are relative, it can be argued that the aqueduct is dated to the late 13th century, although it is unclear if it falls within the Andalusi or the Christian period.

## 7. Certainties and Hypotheses

The research undertaken over the last five years, based on the re-assessment of the written sources and two excavation seasons in 2018 and 2019, has considerably increased our understanding of the estate of Monteagudo. Below, the new data and the hypotheses to which it leads will be briefly presented.

The 11th-century fortress of Monteagudo was rebuilt and enlarged by Ibn Mardanīš as part of his new princely estate, as a defensive reference point and as a state grain depot. The new fortress, which stands tall in the middle of the Murcian plain, singles the estate out from other similar properties and explains why the *almunia* was known in the Arabic sources as Ḥiṣn al-Faraj. The toponym *Qaṣr Ibn Saʿd* (Ibn Mardanīš's palace), which also features in the written record, refers exclusively to the Castillejo, i.e., the residential building constructed on a mound.

The careful review of the book of the *Repartimiento* has yielded more precise data about the size of the estate, which was divided into at least two parts, granted to the queen and the *partidor mayor*. The sum of both grants exceeded 1000 *tahúllas*, which is more than 120 hectares, and this is in addition to the territory that the Crown likely kept for itself until it was handed over to the Church of Cartagena in 1311. The written record suggests that, as well as the palaces, the estate included irrigated areas, dry land, vineyards, woodland, hunting grounds, and marshland.

The need to control the wadis in order to protect the orchards in the plain from flooding triggered the construction of dams to divert rainwater to the rain-fed crops in the higher mountain areas. Exceptionally, the old watercourses were even obliterated, as illustrated by the palatine area in the plain, where a major 12th-century cross-shaped garden was built upon the tail-end of the Caracol wadi.

The main result of the archaeological excavations is the discovery of the palatine area in the plain, which has resulted in a better understanding of the specific function of the Castillejo and the complex integration of each of the architectural elements in an ascending aulic scenography, favoured by the natural orography of the site. The great transept garden, in addition to organising the surrounding architectural features, channelled visual perspectives along its main axis (east-west) and the pavilion towards the Castillejo, built on a promontory as a great lookout point and expression of Ibn Mardanīš's political power, a scenic complement to the neighbouring fortress of Monteagudo.

The large size of the palace complex and its monumental character are a clear expression of the representational needs of an emirate, such as Ibn Mardanīš's, the political rival of one of the most powerful empires of the Middle Ages, the Almohad Caliphate. The importance and significance of the Monteagudo estate, as the highest expression of Ibn Mardanīš's power, is attested to in Ibn Ṣāḥib al-Ṣalāt's chronicle, while the damage inflicted on it by the Almohads in their two sieges of Murcia was also regarded as a major event.

The closest precedents for the palatine area of Monteagudo date to the period of the Ummayad Caliphate. First, the High Garden of in Madīnat al-Zahrā', where a representational building (the hall of 'Abd al-Rahman III) opened onto the transept garden. Second, the *almunia* of al-Rummaniyya only differs from the Monteagudo complex in that the Cordovan pavilion and its pool open onto a terraced garden because the steep slope of the foothills on which it is located precluded a cross-like arrangement. Third, the coeval Cuba Soprana in Palermo's nymphaeum belvedere opens onto a transept garden (Navarro Palazón et al. forthcoming).

The palatine area in the plain, which comprised a complex of buildings arranged in three blocks, was preside by an oblong hall preceded by a portico and a cistern and opening onto a large transept garden. A prototypical example of this typological juxtaposition can be found in the palace enclosure of Dār al-Hana, within the Agdal estate in Marrakesh (Navarro Palazón et al. 2013, 2018a). Another less-known example was discovered 5 km from the city of Murcia, in Santa Catalina del Monte (Verdolay) (Jiménez Castillo 2013, pp. 337–42). It is a hall preceded by a portico and a cistern open to the landscape, located on a narrow hillside with a steep slope; these remains, plus those of a bath, make the identification of the site as an aristocratic estate perfectly plausible. The origin of this model is in the east (Arnold 2009), as shown by the Abbasid examples of Samarrā (9th century) and other distant parallels such as Laškar-i Bazar in Afghanistan (11th century). As noted, the model had already arrived in al-Andalus, as illustrated by the gardens (high and low) of Madīnat al-Zahrā' (10th century) (Vallejo Triano 2010).

In the palace of Monteagudo, the great hall, in addition to the transept garden, opens to a rear courtyard that should be interpreted as the axis of the palace's private area. This model is found in al-Rummaniyya, and Ibn Mardanīš also used it in the Dār aṣ-Ṣuġrà, another of his estates.

The archaeological record, by itself, has not provided sufficiently precise information to date the buildings found. The ceramics, scarce and fragmentary, correspond to two different periods far apart in time: the oldest are Roman, from the first century A.D. and come from the disappeared medieval earthen walls, while the most recent date from the twelfth century and first half of the thirteenth century. When trying to date the buildings, it is necessary to differentiate those of the southern residential area from the pavilion-palace.

In the former, we have identified a complex superposition of structures, in which we distinguish three major architectural phases. The first could be dated to the middle of the 12th century, judging by the construction technique and materials, by the style of the painted baseboards, and by the ceramic materials recovered in the thick layer that documents the abandonment and ruin of the buildings, all consistent with the date of construction of the fortified palace of Castillejo and with the reform of the Castle of Monteagudo. The second phase is represented by the building constructed over the previously mentioned massive stratum and could be dated to the end of the 12th century or the first third of the 13th century, judging by the type of double access opening to the brick-paved hall. The third

and last phase corresponds to the southernmost section of the aqueduct associated with the second residential building; we can only say that its chronology must have been a little later than the second phase in the 13th century, although we cannot say if it is an Andalusi or Castilian work from the third quarter of the 13th century.

In the pavilion-palace we also identified three successive constructive phases that are not entirely coincident with those differentiated in the buildings of the residential area. The first two belong to the foundational palace and to the extension it underwent; the structures were completely razed to levels below the pavements during agricultural work. The ceramic materials collected were found in a very altered stratum of about 40–50 cm that covers the large surface of the building. These two phases are coevals to the first one described for the domestic area and could be dated to the mid-twelfth century. The third phase also corresponds here to the aqueduct, which was built on the ruins of the palace.

In summary, it is thought that both the fortified palace and the pavilion were built around 1150, and that their destruction must have taken place during the first Almohad assault on the estate in 1165. After that date, the former seems never to have been restored, while the latter was rebuilt at the same time that it was endowed with lateral bays. The final destruction of the palatine area in the plain with its great transept must have taken place during the second Almohad raid in 1171, after which the ruins of the pavilion were never rebuilt, as demonstrated by the construction of the aqueduct in the 13th century. The aqueduct may have been the work of Ibn Hūd or the Castilians immediately after the conquest.

**Author Contributions:** These two authors contributed equally to this work. All authors have read and agreed to the published version of the manuscript.

**Funding:** This article has been written in the framework of the ALMEDIMED project "Medieval Almunias in the Mediterranean: History and conservation of periurban cultural landscapes" (PID2019-111508GB-I00), of which Dr. Julio Navarro (EEA-CSIC) is PI. Co-financed with ERDF funds, it belongs to the State Programme for Knowledge Generation and Scientific and Technological Strengthening of the R&D&I System, State Sub-programme for Knowledge Generation, of the Spanish Ministry of Science and Innovation.

**Data Availability Statement:** Data is contained within the article.

**Conflicts of Interest:** The authors declare no conflict of interest.

## Notes

[1] Authors' note: The technical team was composed, in addition to the signatories of this work, of the archaeologists Juan Antonio Ramírez, Francisco Muñoz, and Pablo Cercós; the architects Francisco Javier López and Luis García; the restorer Eva Mendiola; the paleobotanical studies were in charge of Concepción Obón and Javier Valera; the graphic documentation was carried out by Pablo Pineda, José Javier Martínez, Paulina Contreras, and Miguel González; Maurizio Toscano (coordinator of new technologies); Francisco Ramos (administrative coordination) and Inmaculada Camarero (Arabist) also participated. We would like to express our gratitude to all of them and also to our colleagues, Olga and Andy Bush, for their careful review of the manuscript and valuable suggestions.

[2] (Lozano 1794, I, ch. XIX, pp. 160–71; Amador de los Ríos 1889, pp. 526–27; Terrasse 1932, p. 23; Torres Balbás 1933–1934, 1934, 1952; Gómez-Moreno 1951, pp. 279–85; Marçais 1954, p. 214; Navarro Palazón and Jiménez Castillo 1993, 1995b; Manzano Martínez 1998, 2007; Martínez Enamorado et al. 2007; Almagro Gorbea 2008; Almagro Vidal 2008, pp. 225–40; Martínez Enamorado 2009, pp. 225–63; Navarro Palazón and Jiménez Castillo 2012; Robles Fernández 2016a, 2016b; García Granados 2018; Navarro Palazón et al. 2019; Eiroa Rodríguez and Gómez Ródenas 2019).

[3] Some historical studies also mention Ibn Mardanīš's architectural programme, notably Abigail K. Balbale's (2023) monograph.

[4] The building was excavated in the 1980s and again twenty years later by J. A. Martínez López. The results of both excavations remain unpublished, although some of the materials from the first excavation were published in *La cerámica islámica en Murcia* (Navarro Palazón 1986).

[5] In fact, the two palaces destroyed by the Almohad armies in Monteagudo, the Castillejo and the one discovered in 2019, were never restored. Plausibly, the reason for this is that neither the new governors imposed by the 'Unitarians' nor Ibn Hūd, after he became emir, were interested in recovering the memory of Ibn Mardanīš.

6    (Veas Arteseros 1997); from this date onwards, the Monteagudo estate was gradually divided into small lots subject to emphyteutic leases (Rodríguez Llopis 1993, pp. 315–19).

7    The archaeological excavations carried out inside the Alguazas Tower in 1991 revealed a residence with four bays organised around a courtyard (Pujante Martínez 1997). Although the two buildings are technically similar and both have almost square plans and no towers, it should be noted that the one in Alguazas is much smaller (17.90 m sides) than the one in Larache (38 × 40 m). The measurements of the latter exclude the *antemurium*, as it is not known whether the Alguazas Tower had one.

8    The valley is delimited by two parallel mountain ranges, one to the north and the other to the south. The lower-middle section of the southern range is known as Cordillera Prelitoral and comprises the subranges of Carrascoy, Cresta del Gallo, Miravete, Columbares, Altaona, and Escalona, which separate the valley from the wide coastal plain of Campo de Cartagena. The northern range includes the inner rim of the pre-coastal depression, which in its lower sector, already within the Segura depression, comprises lower (no higher than 200 m) and more isolated heights, such as the *cabezos* of Espinardo, El Puntal, Cabezo de Torres, and Monteagudo.

9    Translation by Inmaculada Camarero: (Ibn 'Abdūn 1948, p. 152; 1955, p. 49).

10   Latitude: 38.0438157. Longitude: −1.1223034. UTM: X:664765.80 and Y:4212340.60V.

11   Latitude: 38.0304609. Longitude: −1.1409285. UTM: X:663160.90 and Y:4210825.80V. The Regional Archaeological Charter describes these remains as follows: "*The remains are preserved to a height of 3 m, and a patch at the base appears to have been part of the foundations, which are exposed. It is approximately 1.5 m thick and is made of masonry and lime, with traces of the formwork caissons with which it was built. The feature is undoubtedly medieval and must have been built during the Islamic period, being maintained during the medieval Christian period after the conquest.*".

12   https://listaroja.hispanianostra.org/ficha/castellar-de-churra/ (accessed on 6 March 2023).

13   According to Almela Legorburu's (2015, p. 44) work, the marshlands in the Murcian plain were the unintended result of the accumulation of surplus irrigation water after the Andalusian hydraulic network came into operation. We think, in contrast, that they formed naturally as a result of the endorheic nature of the basin. However, in his chapter about the marshlands of Monteagudo, Almela presents valuable documentation and plans concerning their hypothetical extension and evolution over time (pp. 44–50).

14   It should be noted that some of the elements described in the *Repartimiento*, dated shortly after the conquest of 1266, may pertain to the period of Castilian protectorate (1243–1264), so we should only ascribe them to the Andalusi period with extreme caution. However, from an agricultural point of view, the period of the protectorate must be regarded as Andalusi because it was not until 1257 that the Christians were granted land, and this was limited to a few plots in the Condominas.

15   Torrential runoffs that erode the soil and wash away large volumes of silt and mud.

16   Latitude: 38.0066865. Longitude: −1.1699944. UTM X:660661.70 and Y:4208137.10V.

17   A.M.M., A.C. 1445-6, 1445-XII-11, f.44r in (Molina Molina 1989).

18   A.M.M. Leg. 3.076, No. 6 in Molina Molina (1989): doc. 76.

19   "la Rambla de Alabrache que viene al Caracol" ("the Alabrache Wadi, which comes to Caracol") A.M.M. Leg. 3.076, No. 6 in (Molina Molina 1989, p. 324).

20   It should be noted that, unlike in Huerto Hondo, no remains of interior plastering have been documented in this instance; in fact, none of the three walls preserves its interior face. Neither is there any evidence for pavement, drains, or supply systems. Therefore, it has been identified as a cistern chiefly because it is difficult to interpret it as anything else. We cannot rule out the possibility that it was supplied with water extracted from the water table or that its purposes were purely recreational.

21   The name appears for the first time in a text granting the queen's property to Gonzalo Martínez in 1391 (Torres Fontes 1975, p. 72).

22   1266 (Torres Fontes 1963, p. 31).

23   1279 (Torres Fontes 1963, p. 64).

24   We are grateful to Dr. Inmaculada Camarero for her help with the Arabic sources and the revision of the transliteration of the Arabic words not included in the *Diccionario de la RAE*, for which we have followed the system recommended by the journal *Al-Qanṭara*, except for commonly-used demonyms and city names, in which we have decided to use their Spanish version.

25   Ibn Ṣāḥib al-Ṣalāt's use of the term *basāiṭ* in this paragraph has previously been interpreted as "plain" (e.g., Huici), which is certainly one of its meanings but does not accurately convey the distinction that the author is making between orchards (*basātīn*) and fields dedicated to dryland agriculture (*basā'iṭ*), both of which were found on the plain. For this reason, we have preferred to translate *basāiṭ* as "dry (agricultural) fields".

26   He was born in Cartagena (Murcia) in 1211/2 and was initially trained by his father, who was *cadi* in Cartagena. Afterwards, he completed his studies with other teachers in Murcia, Granada, and Seville. After spending a short time in Morocco, he took up residence in Tunis in 1242, where he died in 1285.

27   Translated by Robert Pocklington (2018).

28   A monument to the Sacred Heart of Jesus was erected on it in 1926 and blown up in 1936; a new monument was inaugurated in 1951.

[29]   The term *faraj*, from the root FRJ, "to comfort, entertain, relax", can be translated in this context into "solace", which expresses one of the functions of these estates.

[30]   The two palaces identified as Mardanisid (the Castillejo and the one in the plain) were destroyed by the Almohads and never restored.

[31]   A series of 16th-century engravings of Tunis and its surroundings includes one by Agostino Veneciano that represents three large suburban royal estates: El Bardo, Ras-at-Tabia, and Abou-Fihr. The latter, which is believed to be the one labelled "Thiergarten" (menagerie or zoo), includes a palatial building and an enclosure with two roaming deer, one of which is shown grazing and the other one in a rampant position. In 1803, the Spanish traveller Ali Bey (Domingo Badia) described the Semelalia estate in Marrakech and its wild gazelles; these annoyed the gardeners by eating the plants, although "the gardens are large enough for them to cause little or no damage" (Ali Bey el Abbassi, Viajes por Marruecos, Madrid: ed. Salvador Barberá, 1984, p. 328).

[32]   The only archaeological materials excavated by Sobejano to survive are fragments of architectural decorations, especially plasterwork. Their study suggests a homogeneous construction programme consistent with a Mardanisid chronology in the mid-12th century. It is worth noting that the painted baseboards discovered there are contemporary with the earliest construction phase in the Dār aṣ-Ṣuġrà, in Murcia.

[33]   José Antonio Manzano put forth this conclusion in a presentation entitled "El poblamiento musulmán en la huerta de Murcia durante el siglo XIII", given in Monteagudo (Murcia) on 12 January 2019, within the framework of seminar *Expansión agrícola y colonización en al-Andalus (siglos X-XIII). El contexto socioeconómico de las almunias de Monteagudo-Cabezo de Torres (Murcia)*. We are grateful for his permission to mention this presentation.

[34]   After a first visit between 4 May and 22 June 1257 (González Jiménez and Carmona Ruiz 2012, pp. 270–73), King Alfonso returned to Monteagudo between late 1271 and 1272, as attested by the records of the fifth distribution (Torres Fontes 1960a, p. 244) and a letter, addressed to the bishop of Cuenca, dated in the *Real de Monteagudo* on 22 February 1272 (González Jiménez and Carmona Ruiz 2012, p. 444).

[35]   As noted by Torres Fontes (1959, p. 64), in the *Repartimiento*, *albar* referred to land whose irrigation status was imprecise, either because it depended on wadi floods or because it was irrigated by means of waterwheels. Their fiscal value was much lower than that of downright irrigated fields; in consequence, *albar* grants were larger than those concerning irrigated fields.

[36]   E.g., Panel C-4, found in situ during the 1924–1925 excavations (Robles Fernández 2019, p. 64). The same solution is found in a fragment in the National Archaeological Museum (Madrid). See Alfonso Robles (Eiroa Rodríguez and Gómez Ródenas 2019, p. 221, catalogue No. 78) for photograph and description.

[37]   The most interesting baseboard was found in the south hall of the palace. Although some authors date it to the Almoravid period (Robles Fernández 2019, p. 56), an early Mardanisid date contemporary to those discovered by Sobejano at Castillejo seems more likely.

[38]   The excavators of the oratory of the Muslim Alcázar of Murcia under the Church of San Juan de Dios found a fragment of painted baseboard that has remained unpublished and whose stratigraphic context is unknown. Although it was dated by Alfonso Robles to the Almoravid period (Eiroa Rodríguez and Gómez Ródenas 2019, pp. 178–79, catalogue No. 36), an early Mardanisid date seems more likely.

[39]   This is a late technique. It has been shown that this type of brick floor only arrived in Murcia with the Castilian conquest (Navarro Palazón and Jiménez Castillo 2011b, pp. 100–104).

[40]   Triple-span openings are found in Madīnat al-Zahrā' (Dār al-Mulk and the Courtyard of the Pillars); the central body of the *almunia* of al-Rummaniyya in Córdoba; the north wing of the Aljafería in Saragossa; and the citadel of Almería.

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
