# Peer review of "The Wolf King’s Leisure Estate: An Andalusi Agricultural and Palatine Project (Murcia, 12th Century)"

_arts, 2023_

Round 1

Reviewer 1 Report

A brief summary

This article is relevant because it provides unpublished information based on the excavations carried out in 2018 and 2019 to the west of the Castillejo de Monteagudo and on the analysis of the surrounding plots of land. This information is contrasted by the author with written sources. It provides a better understanding of this 12th century palatine residence, built during the Islamic Emirate of Ibn Mardanis (1147-1171) and located in the southeast of the Iberian Peninsula. This study helps to better define the limits of the territory under the control of King Lobo. It defines the plan of the territory controlled by Ibn Mardanis and attributes these possessions to the state, the amir and his family or to members of the aristocracy, always in the visual and scenographic discourse of ostentation of prestige and power, based on the creation of a hydraulic infrastructure that highlights the scientific and cultural progress of this kingdom.

General concept comments
Article:
The subject is dealt with exhaustively and with a good command of the documentary and bibliographical sources. It cites classic works of obligatory reference and others of recent publication. One of the most significant advances of the article is that, contrary to what has been defended until now, that is, that the castle was the palatine residence and protocol space, the 2019 excavation has revealed the existence in the plain of most of the 12th-century court complex, with the same functions. A pavilion has also been excavated in the centre, presiding over a large transept garden. In addition to the above-mentioned functions, therefore, the castle would have served as a watchtower and a projection of the image of power, due to its position overlooking the plain.

Specific comments

- For future studies or expansion of this interesting subject, perhaps these Andalusian hydraulic infrastructures could be related to their survival after the Christian conquest. As the author indicates, Alfonso X resorted to specialised labour for the maintenance of these installations. It is possible that they were Mudejars. This could open up new lines in terms of the transmission of working methods and techniques from one culture to another.
- Perhaps Madinat al-Zahra should be mentioned before Section 7 as a precedent for the existence of a central pavilion overlooking a rectangular transept garden, given the importance of this archaeological site. The existence of a terraced garden separated by walls, reminiscent of the upper and lower garden at Madinat al-Zahra, should also be mentioned before this paragraph.
- Page 24, paragraph 4: According to tradition, a possible zoo at Madinat al-Zahra (?)

- Page 9, paragraph 1, line 18. Delete: 443).

- Page 9, paragraph 5, line 11. "mortar walls,;".

- Page 11, paragraph 3: in Spanish "tahúlla", instead of "tahulla".

- Page 15, line 1: ":.18"

- Page 18, paragraph 3, line 9: "qanats" (in italics) and indicate English word.

The work is very well illustrated, with abundant plans, drawings and photographs, all of which are necessary to understand the complexity of this monumental complex.

Author Response

Dear reviewer,
thank you very much for your interest and valuable input. We have corrected the errata you have pointed out and, indeed, we have included the reference to the zoological park of Madinat al-Zahra, which we had forgotten to mention. We also thank you for your suggestion regarding the need to study the survival of hydraulic structures after the Christian conquest, especially in that immediate phase that we could call "Mudejar"; indeed, this is a line of research that should be implemented in the future.
Thanks again

Reviewer 2 Report

I thought this was close to perfect until the end. It is an admirable piece of interdisciplinary research, using geography, written sources and archaeology very effectively - and I would add that it is very rare to do so. It is a very important contribution to knowledge; outside Córdoba, the site has no parallels known to me. But, I discovered at the end, it has one major flaw, which must be corrected: it gives no grounds for its dating to the reign of Ibn Mardanish. I guess that this is on the bases of ceramics, maybe coins? The authors (it is clear that there is more than one) must explain the basis of their dating. (If it is only on the basis of the written sources, they must admit it, and then be much more circumspect about the dates, as the sources are fairly sketchy.)

I have a few minor points, almost all concerning the Arabic.

First, the authors do not distinguish between 'ayn and hamza; they just use an inverted comma.

Qaṣr does not mean 'palace' - the authors know that it refers to the Castillejo, but it is still a castle, not a palace, word.

Rahal means a village in Sicily (see Annliese Nef's book). Clearly it does not in Spain, and I would accept Guichard's translation, as the authors imply too, but it really cannot mean sheepfold.

In an article in English, I would advocate the English, not the Spanish, transliteration of Arabic. In particular, English readers will not know that ŷ is English j, and j is English kh.

Tremecén is Tlemcen in English.

That's all: detail, except for the first, major, point.

Author Response

Dear reviewer,

Thank you very much for your careful reading of our manuscript and for your valuable contributions, which have undoubtedly contributed to substantially improve the text.

In accordance with your suggestions, we have corrected the Arabic transcriptions by adapting them to the Anglo-Saxon system.

We have also added 4 new paragraphs at the end of the article in which we summarize all the data relating to the chronology.

Regarding the term "rahal", it is true that in some contexts it seems to designate realities analogous to the farmhouses (qarya, daya), but we believe that it is a survival, in toponymy, of a different previous reality. That is to say, they were originally rahales (private estates), which ended up being transformed into villages while retaining the old name. This phenomenon happened in al-Andalus; so that even before the Christian conquest, the existence of villages whose toponym reflected a previous reality in which they were private estates (rahal, almunia) is already proven. This was the case, for example, with Raffal Abenayçam (Jiménez, 2018, 760-763) or Munyat Faŷŷ al-Bušra; the case of the latter is particularly interesting because this almunia had already been transformed into a farmstead in the time of Abd al-Rahman II, as Ibn al-Abbar explains: "And the amir ‛Abd al-Raḥmān took her on some of his raids [through the North] but she fell ill and back to Cordoba, reaching the Munyat of Faŷŷ-al-Busšraʿ, near Toledo, she died and was buried there and her tomb became famous because Amir Muḥammad released the people of that farmstead (ahl tilka al-qarya), in his state (dawla), from the magārim or extralegal tax so that they would guard [the tomb] and respect her name."

In Sicily, the written documentation that is handled in relation to the rahales of the region of Monreale is from the end of the s. XII century, that is, a century after the Norman conquest; nevertheless, from it Annelise Nef writes "Des indices suggèrent en effet le prolongement à la période normande d' une pratique antérieure proche de la concession fiscale existant en Islam, l' iqṭāʿ, une situation que semble confirmer, au-delà du Monréalais, the definition of the "fief première manière" in Sicily, identified at just title to an "alleu en terre fiscale," combining the large jouissance of the beneficiary and the possible intervention of the State at any moment. Il est désigné en arabe par le terme de "raḥal" qui renvoie moins à un type d' habitat, comme on l' a longtemps cru, qu' à un type de concession." (Ney and Prigent, 2019, https://www.jstor.org/stable/10.1163/j.ctv2gjwnz5.13) In other words, it would be a type of property analogous to the Andalusian one.

Regarding the term qaṣr; it is true that it has the meaning of "fortress" but also that of "fortified palace" or simply "palace". As can be seen in the main Arabic dictionaries (Kazimirski, vol. II, p. 751; Corriente, p. 627; Lane, vol. 7, p. 2534). According to Van Staëvel in the Encyclopaedia of Islam, in the Maghreb qaṣr has "several semantic layers to be illuminated by careful examination of the various texts available and by archaeological research, and in the light of the complex material factors relating to the Maghrebi habitat from medieval times until much later. Hence, it denotes here: (a) a palace, the place from which political authority is exercised, or an aristocratic residence; (b) a fortified place, a small fortress or a large-scale fortress; (c) a fortified complex for community habitation; and (d) a collective granary or storehouse ". In al-Andalus, moreover, we know of numerous palaces that were called qasr and that can in no way be identified with a fortification, such as the Qaṣr al-Sayyed (Alcazar del Genil in Granada); Qusūr al-Buḥayra (the Palaces of the Buhayra in Seville); Qaṣr al-Sulṭān (Palace of Comares) or Qaṣr el-Riyāḍ el-Saʿīd (Palace of the Patio de los Leones).

Reviewer 3 Report

This article is an important contribution to our knowledge of the construction work carried out by the emir Ibn Mardanis to the north of the city of Murcia. The analysis of the written documentation, the traditional historiography and the archaeological results obtained by the author(s) bring us closer to a better understanding of the surroundings of Qasr b. Sa'd and Hisn al-Muntaqud and to advance in the function that this palace complex played. However, given the importance of this research in the advancement of research, we will have to wait for future studies to shed more light on the subject. I would like to take this opportunity to congratulate the authors for this article and the complete bibliography used for this purpose.

Author Response

Dear reviewer,
thank you very much for your interest and your very positive assessment of our work. Indeed, we hope that this article will serve as a basis for future archaeological research in the Castillejo area to continue documenting this unique aulic complex.
Thank you again